# Galectin-9 interacts with PD-1 and TIM-3 to regulate T cell death and is a target for cancer immunotherapy

Riyao Yang [1,9,10✉], Linlin Sun[1,2,9], Ching-Fei Li[1], Yu-Han Wang[1,3], Jun Yao[1], Hui Li[1,4], Meisi Yan[1,5], Wei-Chao Chang[3], Jung-Mao Hsu[1,3], Jong-Ho Cha[1,6], Jennifer L. Hsu[1], Cheng-Wei Chou[1,3,7], Xian Sun[1,8], Yalan Deng[1], Chao-Kai Chou[1], Dihua Yu[1] & Mien-Chie Hung [1,3,10✉]

The two T cell inhibitory receptors PD-1 and TIM-3 are co-expressed during exhausted T cell differentiation, and recent evidence suggests that their crosstalk regulates T cell exhaustion and immunotherapy efficacy; however, the molecular mechanism is unclear. Here we show that PD-1 contributes to the persistence of PD-1[+]TIM-3[+] T cells by binding to the TIM-3 ligand galectin-9 (Gal-9) and attenuates Gal-9/TIM-3-induced cell death. Anti-Gal-9 therapy selectively expands intratumoral TIM-3[+] cytotoxic CD8 T cells and immunosuppressive regulatory T cells (T$_{reg}$ cells). The combination of anti-Gal-9 and an agonistic antibody to the co-stimulatory receptor GITR (glucocorticoid-induced tumor necrosis factor receptor-related protein) that depletes T$_{reg}$ cells induces synergistic antitumor activity. Gal-9 expression and secretion are promoted by interferon β and γ, and high Gal-9 expression correlates with poor prognosis in multiple human cancers. Our work uncovers a function for PD-1 in exhausted T cell survival and suggests Gal-9 as a promising target for immunotherapy.

[1] Department of Molecular and Cellular Oncology, The University of Texas MD Anderson Cancer Center, Houston, TX, USA. [2] Tianjin Key Laboratory of Lung Cancer Metastasis and Tumor Microenvironment, Lung Cancer Institute, Tianjin Medical University General Hospital, Tianjin, China. [3] Graduate Institute of Biomedical Sciences and Center for Molecular Medicine, China Medical University, Taichung, Taiwan. [4] Department of Liver Surgery and Transplantation, Liver Cancer Institute, Zhongshan Hospital, Fudan University and Key Laboratory of Carcinogenesis and Cancer Invasion, Ministry of Education, Shanghai, China. [5] Department of Pathology, Harbin Medical University, Harbin, China. [6] Department of Biomedical Sciences, College of Medicine, Inha University, Incheon, Korea. [7] Division of Hematology/Medical Oncology, Department of Internal Medicine, Taichung Veterans General Hospital, Taichung, Taiwan. [8] Department of Medical Oncology, The Seventh Affiliated Hospital, Sun Yat—Sen University, Shenzhen, China. [9]These authors contributed equally: Riyao Yang, Linlin Sun. [10]These authors jointly supervised this work: Riyao Yang, Mien-Chie Hung. ✉email: ryang@mdanderson.org; mhung@cmu.edu.tw

C D8 T cells kill tumor cells directly during antitumor immune response, and tumor infiltration of CD8 T cells positively correlates with patient prognosis in a wide range of malignancies[1]. The frequency and activity of these cells are regulated by a balance of co-stimulatory and co-inhibitory signals that are collectively termed immune checkpoints[2]. Under normal physiological conditions, immune response is kept in check by immune checkpoints to prevent autoimmunity. However, persistent antigen stimulation in chronic viral infection and cancer leads to T cell exhaustion, a state of T cell dysfunction regulated primarily by the master regulators TOX[3] and TCF-1[4,5], and developed after precursor exhausted T cells (TCF1$^+$PD-1$^{int}$TIM-3$^-$) differentiate into terminally exhausted T cells (TCF1$^-$PD-1$^{hi}$TIM-3$^+$)[6–10]. The programmed cell death protein 1 (PD-1) is one of the most studied immune checkpoints that regulates T cell activity. Upon engagement by its ligand PD-L1, PD-1 recruits the SHP2 tyrosine phosphatase to dephosphorylate critical protein molecules for TCR signaling and T cell activation, such as ZAP-70 and the co-stimulatory receptor CD28[11,12], which is also the target of the cytotoxic T-lymphocyte antigen-4 (CTLA-4) receptor[13]. Blockade of immune checkpoints by antagonistic antibodies targeting PD-1 or its ligand PD-L1 and CTLA-4 has revolutionized cancer therapy by promoting antitumor immunity[14,15]. Despite impressive successes, many cancer patients still do not benefit from current immune checkpoint therapies. In this study, we sought to identify novel immune checkpoint pathways and mechanisms that can be targeted for cancer immunotherapy.

Immune checkpoint molecules can be regulated at multiple levels. For instance, in addition to transcriptional regulation, post-translational modifications of PD-1[16] and PD-L1[17] have been shown to regulate their protein stability and interaction. The turnover of PD-1 is also regulated by its interaction with the E3 ubiquitin ligase FBXO38[18]. Recently, CD80, the ligand for both CD28 and CTLA-4, was found to interact with PD-L1 in *cis* on antigen-presenting cells (APCs) to repress the PD-1 and CTLA-4 co-inhibitory pathways while promoting CD28 co-stimulation[19,20]. Those findings reveal interesting crosstalk between immune checkpoint pathways in regulating T cell activity.

Galectin-9 (Gal-9) is a member of the galectin family of animal lectins with conserved carbohydrate-recognition domains (CRDs) for β-galactosides[21]. Structurally, Gal-9 consists of two CRDs connected by a linker sequence, and is able to crosslink glycoproteins to form multivalent galectin-glycoprotein lattices that regulate multiple cellular processes, including TIM-3-mediated T cell death[21,22]. Consistent with TIM-3 as a mediator of Gal-9-induced T cell death, terminally exhausted T cells (PD-1$^+$TIM-3$^+$) have reduced long-term survival compared with precursor exhausted T cells that also express PD-1 but lack TIM-3 expression[6]. Nevertheless, PD-1$^+$ TIM-3$^+$ T cells persist in the tumor microenvironment (TME), and even dominate the tumor-infiltrating CD8 T cell pool in some cancer types; their presence is associated with tumor reactivity and predicts response to PD-1 blockade in cancer patients[23–26].

In this work we reveal a molecular mechanism of exhausted T cell persistence, namely, PD-1 physically interacts with Gal-9 and TIM-3 to attenuate Gal-9/TIM-3-induced T cell apoptosis. In addition, we show upregulation of Gal-9 expression and secretion by interferons, identify tumor-infiltrating immune cells that respond to Gal-9 blockade, and devise an effective combination therapy that boosts the antitumor effect of anti-Gal-9 therapy. Altogether, the current work unravels a role for PD-1 in the regulation of exhausted T cell survival, and establishes Gal-9 as an important regulator of tumor immune response that can be targeted for cancer immunotherapy.

## Results

**Galectin-9 is a PD-1-binding protein.** To further understand the function of PD-1, we sought to identify additional PD-1-binding proteins by expressing a C-terminal 3×FLAG-tagged PD-1 (PD-1.3F) in Jurkat T cells using a doxycycline-inducible retro-lenti-viral system[27] followed by immunoprecipitation (IP) with a FLAG antibody. Mass spectrometric analysis of the immuno-complexes identified Gal-9 as a major binding protein for PD-1 with known immunomodulatory activity (Supplementary Fig. 1a). This was further validated by IP/Western (Fig. 1a) and pulldown assays with Gal-9 immobilized on Sepharose 4B beads (Fig. 1b). Because Gal-9 is an animal lectin and PD-1 a glyco-protein (Supplementary Fig. 1b), their binding is likely glycan-mediated. Indeed, the addition of lactose, an inhibitor of galectin-glycan interactions, but not sucrose, blocked the binding of Gal-9 to PD-1 (Fig. 1b). To show that the binding is direct and specific, we utilized a plate-based binding assay (Fig. 1c) with purified Fc-fusion proteins of the extracellular domain (ECD) of potential Gal-9-binding proteins and found significant binding of Gal-9 to PD-1 and TIM-3, but not to PD-L1 (Fig. 1d). Other galectins, such as Gal-8 or Gal-1, did not exhibit PD-1-binding activity (Supplementary Fig. 1c). Gal-9 and PD-L1 did not compete with each other for PD-1 binding (Supplementary Fig. 1d, e). In addition, pembrolizumab and nivolumab, the two FDA-approved therapeutic monoclonal antibodies against PD-1, decreased PD-1/PD-L1 binding but not PD-1/Gal-9 binding (Fig. 1e). Thus, the data suggested that the binding sites on PD-1 for Gal-9 are distinct from those for PD-L1 and the two therapeutic antibodies. Together, these results indicated that Gal-9 binding to PD-1 is highly selective and mediated by glycans, and it does not affect PD-1 binding to its cognate ligand PD-L1 or the PD-1 therapeutic antibodies pembrolizumab and nivolumab.

**Binding of Gal-9 to PD-1 is primarily mediated by the C-terminal carbohydrate-recognition domain (CRD) of Gal-9 and the N116-linked glycan of PD-1.** Gal-9 consists of two CRDs, N-terminal CRD (N-CRD) and C-terminal CRD (C-CRD), with similar but distinct specificities for glycans[21,28]. To determine which CRD mediates Gal-9 binding to PD-1, we purified the two CRDs individually as GST-fusion proteins (Fig. 2a) and found that GST-9C (C-CRD) exhibited greater binding to PD-1 compared to GST-9N (N-CRD) (Fig. 2b and Supplementary Fig. 1e), whereas the two CRDs exhibited largely equal binding to TIM-3. This was further corroborated by binding assays using Gal-9 mutants with loss-of-function point mutations in the N-CRDs (R65A) and the C-CRD (R239A), respectively[29]. As shown in Fig. 2c, the two mutants retained significant and largely equal TIM-3-binding activity. In contrast, significant PD-1-binding activity was lost when either domain of Gal-9 was mutated (compare Fig. 2c with Fig. 1d); the loss of PD-1 binding activity was more severe for the C-CRD (R239A) mutant compared with the N-CRD (R65A) mutant (Fig. 2c). The results suggest that Gal-9 binds to PD-1 primarily through its C-CRD, whereas both N-CRD and C-CRD mediates its binding to TIM-3. To determine the glycosylation sites on PD-1 contributing to Gal-9 binding, we mutated the asparagine (N) residues in the four putative glycosylation sites (N49, N58, N74, and N116) to glutamine (Q) individually and assessed the effects on Gal-9 binding. We found that binding of PD-1 to Gal-9 was largely abolished by the N116Q mutation, although binding was also moderately reduced by the other three mutations (Fig. 2d). We

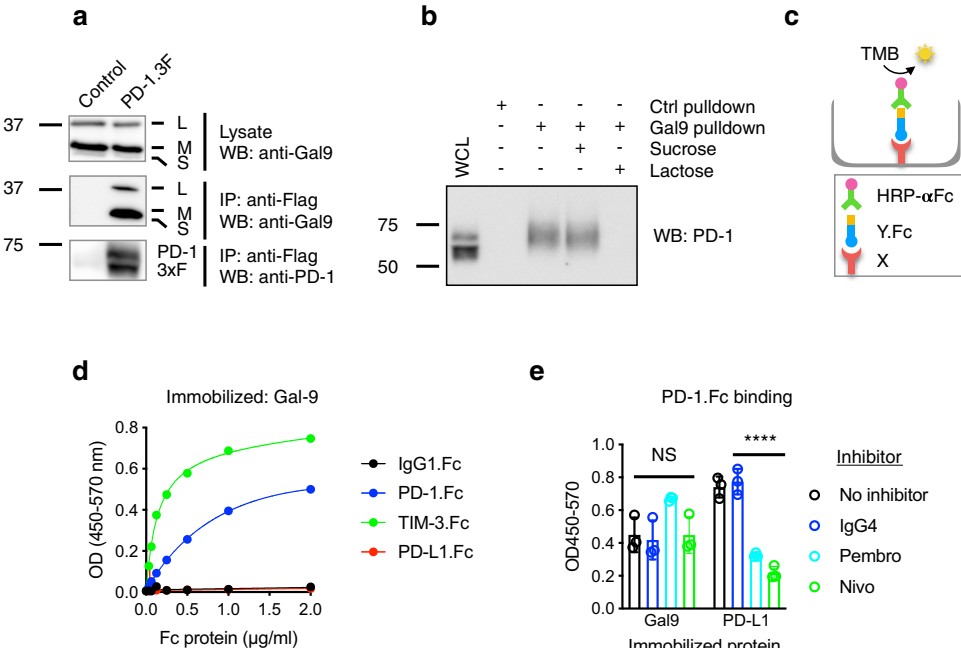

**Fig. 1 Galectin-9 is a PD-1-binding protein. a** Lysates of Jurkat cells transduced with control lentivirus or PD-1 tagged at the C-terminus with 3× FLAG tag (PD-1.3F) were immunoprecipitated with anti-FLAG magnetic beads and the associated proteins were subjected to immunoblotting with Gal-9 or PD-1 antibodies. The three Gal-9 bands (L, M, S) represent different isoforms resulted from alternative pre-mRNA splicing. **b** Jurkat PD-1 cell lysates were incubated with glutathione-Sepharose (control) or Gal-9-Sepharose beads with or without sucrose or lactose. Bound proteins were eluted and western blotted with anti-PD-1 antibody. **c, d** Plate-based binding assay with purified recombinant proteins shows direct and specific binding of PD-1 extracellular domain (ECD) to Gal-9. MaxiSorp plates were coated with Gal-9 and incubated with Fc-fusion protein of the ECD of test binding partners (PD-1, PD-L1, or TIM-3) at various concentrations. Binding was detected by spectrophotometry using an HRP-labeled anti-human IgG (Fc-specific) antibody and the HRP substrate TMB (3,3′,5,5′-tetramethylbenzidine). X: protein immobilized on plate; Y.Fc: Fc conjugated potential binding protein or IgG1-Fc (control); HRP-αFc: HRP (horseradish peroxidase)-labeled anti-Fc antibody. **e** Binding of PD-1 ECD to immobilized Gal-9 or PD-L1 ECD in the absence or presence of the PD-1 antibodies pembrolizumab (Pembro) or nivolumab (Nivo). $n = 3$ independent experiments. Error bars represent SD. Statistical differences were assessed using two-way ANOVA with Sidak's multiple comparisons test. IgG4 vs Nivo, $P < 0.0001$; IgG4 vs Pembro, $P < 0.0001$. Data are representative of three (**a**, **b**) or two (**d**) independent experiments. Source data are provided as a Source data file.

concluded that Gal-9/PD-1 interaction is primarily mediated by the C-CRD of Gal-9 and the N116-linked glycan of PD-1.

**Gal-9 crosslinks PD-1 and TIM-3 to form galectin/glycoprotein lattices.** Preferential binding of PD-1 to the C-CRD of Gal-9 suggests that it could compete with TIM-3 for C-CRD binding to form TIM-3/Gal-9/PD-1 complexes. Indeed, we found that the PD-1 ECD efficiently competed with TIM-3 ECD for binding to Gal-9 C-CRD (Ki = 20.47 nM) (Fig. 3a). By contrast, PD-1 ECD at concentrations up to 10 μg/ml did not compete with TIM-3 ECD for binding to Gal-9N; at low concentrations it even promoted TIM-3 ECD binding to Gal-9N (Fig. 3b). Such competition is predicted to lead to the formation of TIM-3/Gal-9/PD-1 complexes. Indeed, we showed that PD-1 ECD did not bind TIM-3 ECD in the absence of Gal-9 (Fig. 3c), and Gal-9 promoted their cooperative binding, as indicated by the sigmoidal binding curve (Fig. 3d). This was further corroborated by results from DuoLink assay (Fig. 3e) and IP/Western (Fig. 3f, lanes 8 vs. 4) with cells co-expressing these two receptors. Interestingly, PD-1/TIM-3 interaction was also detected in the absence of exogenous Gal-9 (Fig. 3f, lane 4), and such interaction was not inhibited by lactose (Fig. 3g) or by the N116Q mutation that abolishes Gal-9 binding (Fig. 3h). These results suggest that the two receptors can also interact with each other in a Gal-9-independent manner, likely through their intracellular domains (ICDs). The addition of Gal-9 to cells expressing TIM-3 partitioned a portion of the two proteins to the pellet fraction of the cell lysate (Fig. 3i), indicating the formation of insoluble cross-linked galectin-glycoprotein

lattices[30]. In accordance with the observation that Gal-9 mainly uses its C-CRD to interact with PD-1 (Fig. 2b, c) and thus could not efficiently crosslink the PD-1 molecules, PD-1 did not form insoluble lattices with Gal-9 in the absence of TIM-3 (Fig. 3j). However, when Gal-9 was added to cells co-expressing PD-1 and TIM-3, all three molecules were found in the pellet fraction (Fig. 3k), consistent with the formation of insoluble (TIM-3/Gal-9/PD-1)$_n$ tri-molecular lattices. Taken together, these data suggest that PD-1 and TIM-3 form heterodimers via their intracellular domains, and Gal-9 crosslinks these dimers to form galectin-glycoprotein lattices (Fig. 3l).

**Co-expressed PD-1 protects TIM-3⁺ T cells from Gal-9-induced cell death.** We next investigated the functional significance of PD-1/Gal-9 interaction. Gal-9 and TIM-3 interaction was initially reported to suppress type 1 helper T cell (CD4⁺) immunity[22] but was later shown to also dampen CD8⁺ T cell response[31]. We found that although both CD4⁺ and CD8⁺ T cells were sensitive to Gal-9-induced cell death, CD8⁺ cytotoxic T cells were particularly more so than CD4⁺ T cells (Supplementary Fig. 2a–d). The apoptosis-inducing activity of Gal-9 required both CRDs (Supplementary Fig. 2d and e) even though each CRD by itself was sufficient to bind to TIM-3 (Fig. 2b, c).

Most TIM-3⁺ T cells in tumors co-express PD-1. These PD-1⁺ TIM-3⁺ T cells are supposed to be susceptible to Gal-9-induced apoptosis, yet they persist in the TME and even dominate the intratumoral CD8 T cell population in some mouse and human cancers[24–26]. One possibility is that co-expressed PD-1 inhibits

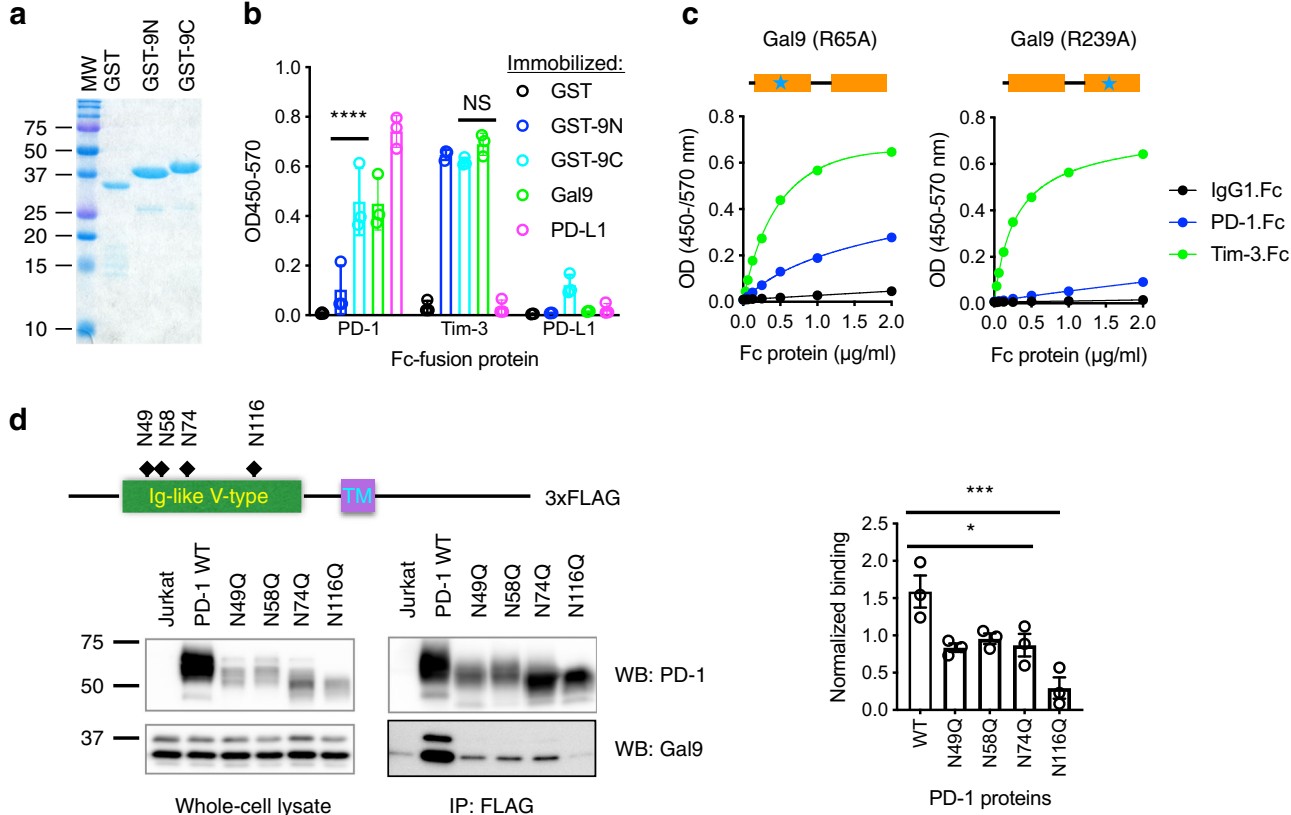

**Fig. 2 Binding of Gal-9 to PD-1 is primarily mediated by the C-CRD of Gal-9 and the N116-linked glycan of PD-1. a** SDS-PAGE of GST and GST-fusion proteins of the N-CRD (GST-9N) and C-CRD (GST-9C) of Gal-9. **b** Plate-based binding assay measuring the binding of PD-1, TIM-3, and PD-L1 to indicated proteins immobilized on MaxiSorp plates. $n = 3$ independent experiments. Error bars represent SD. Statistical differences were assessed using two-way ANOVA with Tukey's multiple comparisons test. ****$P < 0.0001$. **c** Plate-based binding assays of PD-1 and TIM-3 binding to immobilized Gal-9 mutants with loss-of-function point mutation in the N-CRD (R65A) or C-CRD (R239A), respectively. **d** Lysates of Jurkat cells expressing 3xFLAG-tagged WT PD-1 or glycosylation site mutants were incubated anti-FLAG M2 magnetic beads. Bound proteins were eluted and subjected to Western blotting with PD-1 or Gal-9 antibodies. Statistical differences were assessed using ordinary one-way ANOVA with Dunnett's multiple comparisons test. *$P < 0.05$; ***$P < 0.001$. WT vs N49Q, $P = 0.0113$; WT vs N58Q, $P = 0.0302$; WT vs N74Q, $P = 0.0148$; WT vs N116Q, $P = 0.0002$. Data are representative of two (**a**, **c**) or three (**d**) experiments. Source data are provided as a Source data file.

Gal-9/TIM-3-induced T cell apoptosis and contributes to their persistence. To test this, we expressed TIM-3 and PD-1 individually or together in human Jurkat T cells and measured Gal-9-induced apoptosis. We used the dox-inducible expression system, anticipating that constitutive TIM-3 expression would be toxic to cells. Cell survival was determined by viable cell counts of relevant cell populations with or without Gal-9 treatment. As expected, compared with control, TIM-3 expression sensitized cells to Gal-9-induced cell death, which was rescued when wildtype PD-1, but not the Gal-9-binding-deficient N116Q mutant, was co-expressed (Fig. 4a–c). In the absence of TIM-3, wildtype PD-1 (but not the N116Q mutant) moderately inhibited Gal-9-induced apoptosis (Fig. 4b, c). The results suggest that co-expressed PD-1 suppresses Gal-9/TIM-3-induced T cell apoptosis, and such suppression requires its glycan-mediated binding to Gal-9.

To further validate these results in primary T cells, we treated primary CD8 T cells with or without Gal-9 in the presence of ImmunoCult Human CD3/CD28/CD2 T Cell Activator. We found that PD-1+TIM-3+ T cells survived better than PD-1− TIM-3+ T cells after Gal-9 treatment (Fig. 4d, e). There are some reports that Gal-9 can activate T cells[32], which could potentially increase PD-1 expression and complicate the interpretation of the above data. To address this concern, we measured the effects of Gal-9 on PD-1 expression in this assay. We found that under the

above experimental conditions, ImmunoCult Human CD3/ CD28/CD2 T Cell Activator strongly induced PD-1, but Gal-9 did not further substantially increase the percentage of PD-1+ cells (Supplementary Fig. 2f).

Taking together, the data so far are consistent with the notion that Gal-9 promotes TIM-3-mediated T cell apoptosis by crosslinking TIM-3 and facilitating TIM-3 aggregation; co-expressed PD-1 attenuates Gal-9/TIM-3-induced apoptosis by promoting the formation of TIM-3/Gal-9/PD-1 lattices.

**Gal-9 is a target for cancer immunotherapy.** Gal-9-induced T cell death could contribute to suppression of anti-cancer immunity. We, therefore, evaluated the potential of Gal-9 inhibition in cancer therapy. Gal-9 expression is significantly altered in most human cancers; in all the cancer types with altered Gal-9 expression, the Gal-9 gene (*Lgals9*) is overexpressed (Supplementary Fig. 3a). Furthermore, high Gal-9 expression is associated with poor prognosis in multiple human cancers (Supplementary Fig. 3b). In vitro, we found that Gal-9 preferentially induced death of primary T cells whereas leukemic T cells and other tumor cells, which do not express TIM-3, were considerably refractory to Gal-9-induced cell death (Supplementary Fig. 2g, h). Using an in vitro co-culture system of engineered T cell cytotoxicity toward tumor cells (Supplementary Fig. 2i), we showed that Gal-9-induced CD8+ T cell death

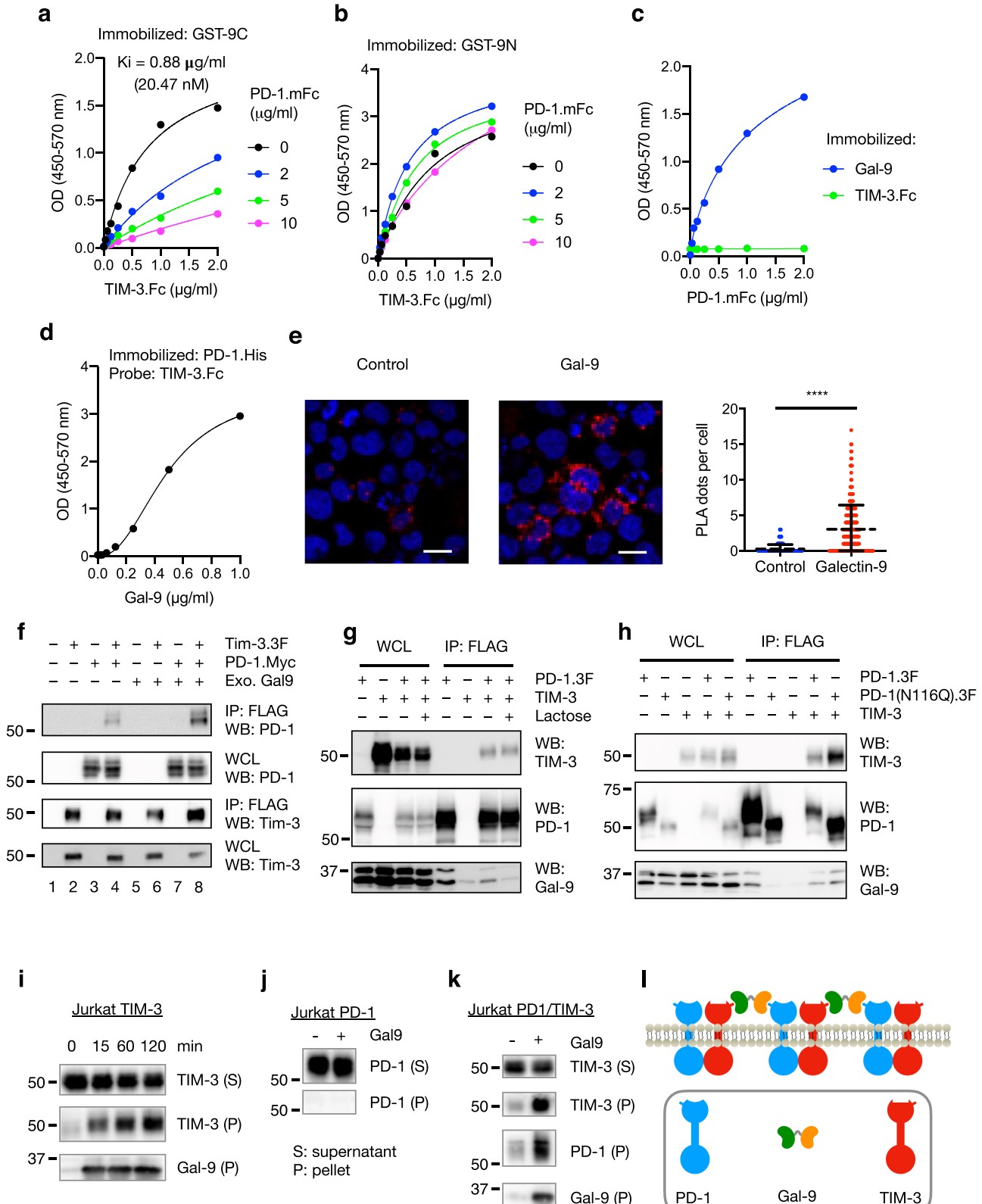

correlated with tumor cell survival (Supplementary Fig. 2j). The data suggested that preferential killing of T cells over cancer cells by Gal-9 may contribute to cancer immune escape.

We predicted that TIM-3+ CD8 T cells, which represent CD8 T cells at functionally distinct intermediate stages of exhausted T cell differentiation[33], can be rescued from Gal-9/TIM-3-induced cell death by Gal-9 inhibition to exert antitumor immunity. A rat

anti-mouse Gal-9 monoclonal antibody, RG9-1, has been shown to inhibit Gal-9 binding to TIM-3 and suppress Gal-9-induced T cell death in vitro[34], and accelerate allograft rejection in vivo[35]. However, RG9-1 monotherapy produced only modest and transient antitumor effects in the MC-38 syngeneic mouse colon cancer model (Fig. 5a–c; Supplementary Fig. 4a, b). We postulated that this may be due to impaired T cell co-

**Fig. 3 Characterization of TIM-3/Gal-9/PD-1 tri-molecular interaction. a, b** TIM-3 ECD binding to plate-immobilized GST-Gal-9C (**a**) or GST-Gal-9N (**b**) in the presence of increasing concentrations of PD-1 ECD. **c** PD-1 ECD binding to plate-immobilized TIM-3 ECD or Gal-9. **d** TIM-3 ECD binding to plate-immobilized PD-1 in the presence of increasing concentrations of Gal-9. **e** Duolink assay of PD-1 and TIM-3 association in Gal-9 KO Jurkat cells co-expressing the two receptors with or without Gal-9. Scale bar: 10 μm. Dashed lines represent mean values; error bars represent SD. Statistical differences were assessed using unpaired two-tailed $t$-tests. $n = 254$ cells examined for each group over two independent experiments. ****$P < 0.0001$. **f** Jurkat cells expressing PD-1 (myc tagged) and TIM-3 (3xFlag tagged) individually or together were incubated with or without 2 μg/ml exogenous Gal-9 followed by IP/western blotting with indicated antibodies. **g, h** IP/Western analysis of Jurkat cells expressing TIM-3 and 3xFlag tagged wildtype PD-1 or PD-1(N116Q) mutant, individually or in indicated combinations, in the presence or absence of lactose. **i–k** Jurkat cells expressing PD-1 (**i**) or TIM-3 (**j**) or both (**k**) were incubated with or without Gal-9, and then lysed in a detergent buffer and centrifuged. Protein levels in the supernatants (S) and pellets (P) were determined by western blotting with the indicated antibodies. **l** Schematic diagram showing TIM-3/Gal-9/PD-1 tri-molecular interactions. TIM-3 and PD-1 dimerize through their intracellular domains. Gal-9 crosslinks TIM-3/PD-1 dimers with its N-CRD (green) and C-CRD (orange) to form galectin/glycoprotein lattices. Data are representative of two (**a–i**) or three (**j, k**) independent experiments. Source data are provided as a Source data file.

stimulation after Gal-9 inhibition, as Gal-9 has been shown to be required for the signaling of 4-1BB[36], a T cell co-stimulatory receptor of the tumor necrosis factor receptor superfamily (TNFRSF). On top of anti-Gal-9-induced T cell survival, compensation for the loss of such co-stimulation may be required to improve the quality of the antitumor immune response. We tested this idea by combining anti-Gal-9 with an agonistic antibody (DTA-1) to GITR (another member of the TNFRSF family of co-stimulatory receptors). DTA-1 has been shown to promote both the clonal expansion of antigen-specific T cells and generation of long-term memory[37,38]. Remarkably, while the antitumor effects of anti-Gal-9 or anti-GITR alone were transient and modest, their combination synergistically suppressed tumor growth and prolonged overall survival (Fig. 5a–c; Supplementary Fig. 4a, b). The effects of the combination were also validated in another syngeneic tumor model with a different strain of mice (BALB/cJ) bearing orthotopic tumors of EMT-6 mouse triple-negative breast cancer (Fig. 5d–f). The dose schedules we used for anti-Gal-9/anti-GITR or their combination did not produce significant antitumor effects in the poorly immunogenic B16 mouse melanoma model (Supplementary Fig. 4c, d). This is not surprising, as this model does not respond well to other immune checkpoint therapies, including PD-1/CTLA-4 blockade either, unless combined with vaccines or chemotherapies that enhance tumor immunogenicity[39,40]. We also examined the therapeutic efficacy of anti-Gal-9/anti-PD-L1 combination therapy in the EMT-6 model. A combination of four doses of anti-Gal-9 followed by four doses of anti-PD-L1 resulted in better survival than monotherapy (Supplementary Fig. 4e–h). Combination of anti-Gal-9 with anti-TIM-3 has not been attempted because the two presumably act on overlapping pathways and their combination is unlikely to yield synergistic effects. Together, these results showed that Gal-9 is a target for cancer immunotherapy, and combination of anti-Gal-9 with GITR agonism induces potent anti-tumor activities.

**Anti-Gal-9 therapy targets specific subsets of tumor-infiltrating T cells.** To identify changes in immune cells that may contribute to the treatment effects, we utilized mass cytometry (CyTOF) to profile the tumor immune infiltrate from each treatment group, using a panel of 30 antibodies against various lineage and functional markers of immune cells (Supplementary Table 1). Unsupervised clustering analysis of CD45+ immune cells from tumors using viSNE[41] and FlowSOM[42] identified 8 major immune cell populations or clusters (Supplementary Fig. 5). Two of the populations exhibited visual differences across the treatment groups (Supplementary Fig. 5, Clusters 5 and 7). We then manually annotated all eight clusters based on marker expression (Supplementary Fig. 6; Fig. 6a). Of those, clusters 5 and 7 represented CD4 T cells and a subset of CD8 T cells (CD8 T_1), respectively (Fig. 6b).

Quantitative analysis indicated an approximately two-fold increase in CD4 T cell frequency (% in total CD45+ cells) in tumors from mice treated with anti-Gal-9 (Fig. 6c, blue vs. black). There was a tendency for CD4 T cell decline in samples from anti-GITR-treated mice although a statistical significance was not reached (Fig. 6c, green vs. black). Interestingly, combination of anti-Gal-9 and anti-GITR led to marked reduction in CD4 T cells (Fig. 6c, red vs black). Many of these intratumoral CD4 T cells from control and anti-Gal-9-treated groups co-expressed FoxP3 and CD25 (Supplementary Fig. 7a), indicating they are $T_{reg}$ cells that suppress CD8 T cell response. It has been reported that Gal-9 enhances the stability and function of $T_{reg}$ cells by interacting with CD44[43], yet we found that similar to conventional CD4 T cells, human $T_{reg}$ cells are susceptible to Gal-9-induced cell death (Supplementary Fig. 7b, c). This suggests that in the TME, Gal-9's $T_{reg}$ killing activity dominates its $T_{reg}$ promoting activity. Tumor-infiltrating $T_{reg}$ cells may be especially susceptible to Gal-9 killing as they express high levels of TIM-3 compared with their counterparts in the periphery[44,45]. Expansion of these cells by anti-Gal-9 led to an increased frequency of $T_{reg}$ cells in total CD45+ TILs in anti-Gal-9-treated mice (3.8% vs. 1.7% in control, Fig. 6d). In line with the findings previously reported[46], intratumoral $T_{reg}$ cell frequency was significantly reduced in mice treated with anti-GITR (Fig. 6d). Remarkably, anti-Gal-9 combined with anti-GITR led to a near-complete loss of $T_{reg}$ cells (Fig. 6d), suggesting that anti-Gal-9-rescued $T_{reg}$ cells are especially vulnerable to GITR-mediated depletion. The mechanism of such synergy is intriguing and remains to be elucidated.

Consistent with the above data that CD8 T cells are sensitive to Gal-9-induced apoptosis (Supplementary Fig. 2), the frequency of the CD8 T_1 subset of CD8 T cells was increased by greater than two-fold after anti-Gal-9 treatment (Fig. 6e). Strikingly, the combination of anti-Gal-9 and anti-GITR further increased the frequency of the CD8 T_1 subset by about 4-fold (Fig. 6e, red vs. black). By contrast, the frequencies of the remaining subset of CD8 T cells (CD8 T_2) and total CD8 T cells was not significantly altered by any of the treatments (Fig. 6e). As a result of decreased $T_{reg}$ cells, the anti-Gal-9/anti-GITR combination induced a high CD8 T/$T_{reg}$ cell ratio (54 for combination vs. 3.6 for anti-Gal-9 alone; Fig. 6f), which was also confirmed by flow cytometry (Supplementary Fig. 7d–f).

As shown in Fig. 6a, compared with CD8 T cells in CD8 T_2 (cluster 8), those in CD8 T_1 (cluster 7) expressed higher levels of T cell activation (CD25, CD69, Ki67) and memory (CD127, CD44, CD62L) markers as well as several T cell co-inhibitory (KLRG1, PD-1, PD-L1, TIM-3, CTLA-4, LAG-3, TIGIT) molecules. Notably, most of CD8 T_1 cells are CD44+CD62L+ (Supplementary Fig. 7g), a central memory T cell ($T_{cm}$) phenotype, as well as PD-1+TIM-3+ (Supplementary Fig. 7h), an exhausted T cell phenotype, and yet are also proliferative, as indicated by Ki67 expression (Supplementary Fig. 7i). Some of the

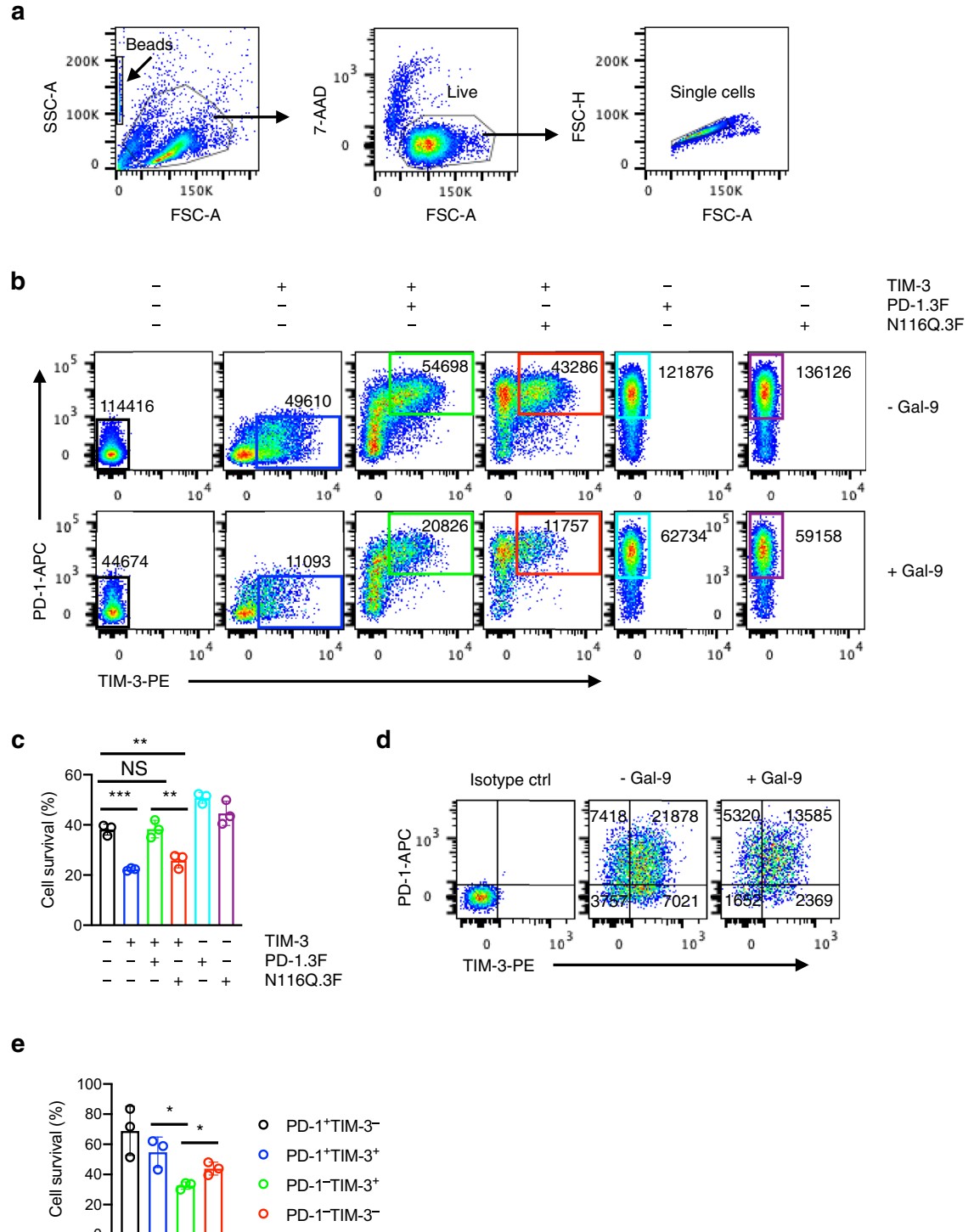

**Fig. 4 Co-expressed PD-1 protects TIM-3+ T cells from Gal-9-induced cell death. a–c** Jurkat cells transduced with indicated proteins individually or in combinations were treated with or without Gal-9 for two days and stained with PD-1/TIM-3 antibodies. Cell survival of relevant PD-1/TIM-3 subsets was determined by flow cytometry with counting beads. **a** Cells were gated based on FSC/SSC parameters and 7-AAD staining. **b, c** Viable single cells equivalent to 3000 counting beads are shown in plots for each sample (10,000 beads were added to each sample just prior to data acquisition). Numbers in plots indicate cell count in corresponding gates. Data (mean values ± SD) from three independent experiments are shown (**c**). Two-tailed unpaired $t$-test. NS, not significant ($P > 0.05$); **$P < 0.01$; ***$P < 0.001$. Control vs TIM-3, $P = 0.0003$; control vs TIM-3 + PD-1, $P = 0.9382$; control vs TIM-3 + PD-1 (N116Q), $P = 0.0040$; TIM-3 + PD-1 vs TIM-3 + PD-1(N116Q), $P = 0.0084$. **d, e** Human CD8 T cells were incubated in ImmunoCult-XF T Cell Expansion Medium with or without Gal-9 in the presence of IL-2 and ImmunoCult Human CD3/CD28/CD2 T Cell Activator for 2 days and analyzed by flow cytometry with counting beads as described above for the survival of different PD-1/TIM-3 subsets. Numbers in plots indicate cell counts in corresponding quadrants. Data (mean values ± SD) are representative of three independent experiments. Two-tailed unpaired $t$-test. *$P < 0.05$. PD-1−TIM-3+ vs PD-1+ TIM-3+, $P = 0.0212$; PD-1−TIM-3+ vs PD-1−TIM-3−, $P = 0.0164$. Source data are provided as a Source data file.

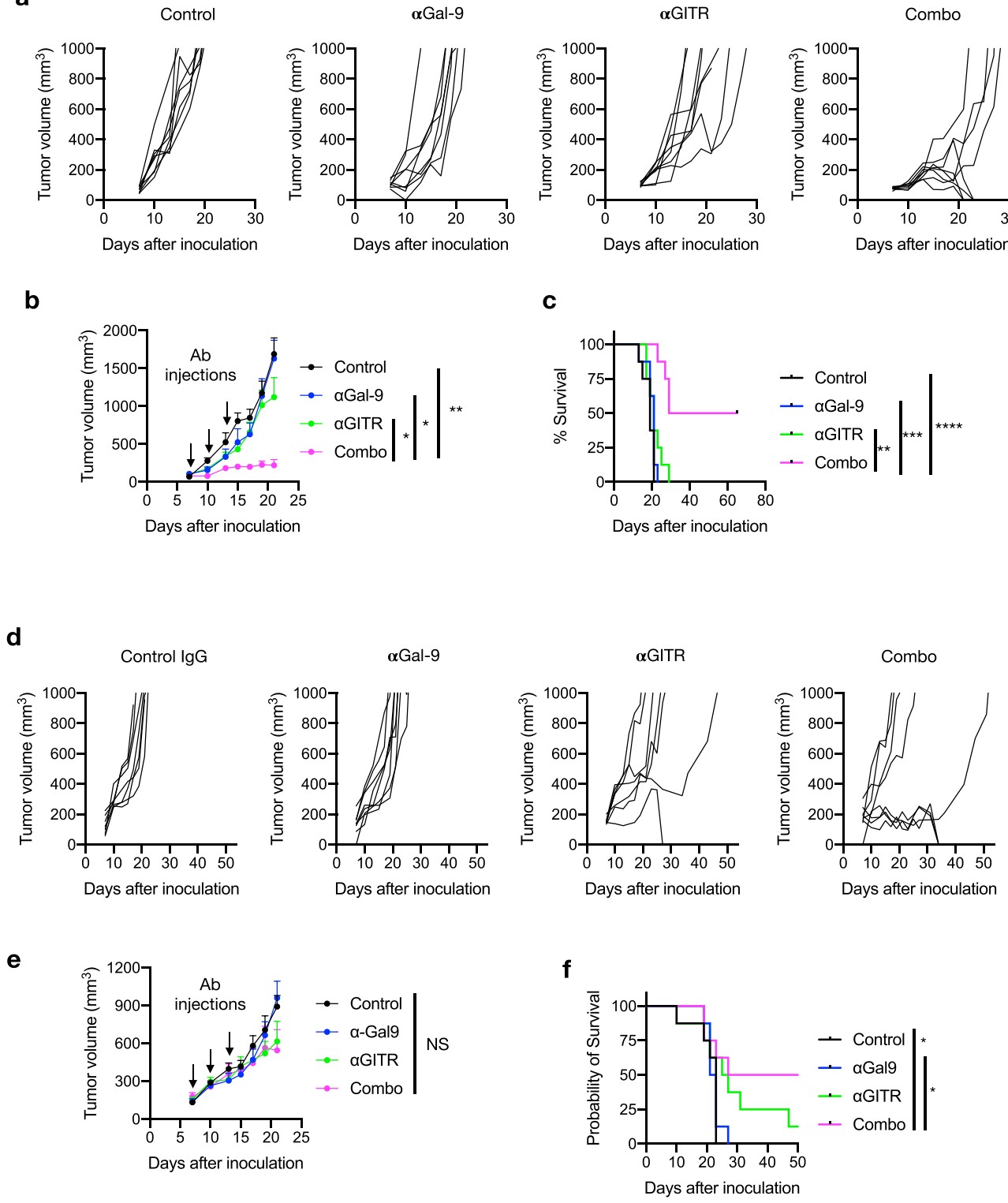

Ki67+ cells also stained weakly positive for cleaved caspase-3 (Supplementary Fig. 7j). This probably reflects T cell activation rather than cell death, as it has been reported that c-Caspase-3 levels are associated with antigen-driven expansion of T cells and not with cell death[47]. Taken together, such mixed phenotype of CD8 T_1 is similar to what was observed in CD8 T cells that respond to other immune checkpoint therapies and are critical for tumor control[6,10,48], and is indicative of transitory T cells in the process of differentiation from precursor exhausted T cells to

terminally exhausted T cells[8,33]. We envisage that Gal-9 inhibition rescues these transitory cells from Gal-9-induced cell death, allowing them to proliferate and differentiate (Fig. 6g). Thus, anti-Gal-9 monotherapy selectively expanded a subset of CD8 T cells with effector potentials. Nevertheless, its therapeutic efficacy is likely compromised by the co-expansion of $T_{reg}$ cells. Indeed, it has been shown that intratumoral TIM-3+ Tregs are especially suppressive[49]. On the other hand, a high frequency of the CD8 T_1 subset and a high CD8 T/$T_{reg}$ ratio in tumors

**Fig. 5 Gal-9 is a target for cancer immunotherapy. a** Tumor growth curves of individual C57BL/6J mice inoculated with MC-38 tumors at day 0 and subjected to indicated treatment. **b** The average tumor growth of mice inoculated with MC-38 tumor cells and subjected to the indicated treatments. Each dot represents mean of 8 mice in each treatment group. Error bars represent SEM of the means. Treatment schedule is indicated by arrows. Statistical differences of tumor growth kinetics between treatment groups were assessed using unpaired two-tailed t-tests to compare area under the curves. Control vs combo, $P = 0.0023$; αGal-9 vs combo, $P = 0.0302$; αGITR vs combo, $P = 0.0192$. **c** Log-rank (Mantel–Cox) tests for comparison of survival curves of mice inoculated with MC-38 tumors. Control vs combo, $P < 0.0001$; αGal-9 vs combo, $P = 0.0001$; αGITR vs combo, $P = 0.0017$. **d** Tumor growth curves of individual BALB/cJ mice inoculated with EMT6 tumors at day 0 and subjected to indicated treatment. **e** Average tumor growth of mice inoculated with EMT6 tumor cells and subjected to the indicated treatments. $n = 8$ mice in each treatment group. Error bars represent SEM of the means. Treatment schedule is indicated by arrows. Statistical differences of tumor growth kinetics between treatment groups were assessed using unpaired two-tailed t-tests to compare area under the curves. **f** Log-rank (Mantel–Cox) tests for comparison of survival curves of mice inoculated with EMT6 tumors. Control vs combo, $P = 0.0363$; αGal-9 vs combo, $P = 0.0433$; control vs αGITR, $P = 0.0666$; αGITR vs combo, $P = 0.2394$. Data are representative of two (**a–c**) or one (**d–f**) independent experiments. Source data are provided as a Source data file.

treated with the anti-Gal-9/anti-GITR combination indicates a strong immune response and likely account for the observed strong antitumor effects.

**Interferon β and γ promote Gal-9 expression and secretion.** Consistent with a major role for Gal-9 in immune response, gene expression analysis of 79 human tissues/cell types indicated that the *LGALS9* gene (encoding Gal-9) was predominately expressed in immune cells, particularly myeloid cells, including dendritic cells and monocytes (Supplementary Fig. 8a. Data from BioGPS). Flow cytometric analysis of human PBMCs showed high Gal-9 expression in monocytes and low expression in T cells (Supplementary Fig. 8b–d). In most cancer cell lines, the expression of Gal-9 was low or undetectable with the exception of some leukemic cell lines, such as Jurkat T cells and THP-1 monocytic cells (Fig. 7a). We then asked whether Gal-9 expression is regulated by immunomodulatory cytokines/factors that are typically present in the TME during an anti-tumor immune response, including IFNβ, IFNγ, and TNFα. We found that IFNβ strongly upregulated the levels of both Gal-9 protein and mRNA in A375 human melanoma cells, whereas IFNγ only moderately induced Gal-9 and TNFα did not have a detectable effect (Fig. 7b). We further tested the effects of IFNβ and IFNγ in additional human and mouse cell lines established from different cancer types, including HCC, prostate cancer, pancreatic cancer and lung cancer, and found that in most cell lines IFNβ consistently and strongly induced Gal-9 expression while IFNγ alone only had a weak or undetectable effect (Fig. 7c–f, Supplementary Fig. 8e–h). Interestingly, while IFNβ strongly induced Gal-9 in human lung cancer cell lines with wildtype EGFR (A549, H441, H1229), it failed to do so in those with mutant EGFR (H1650, H1975, HCC827) (Fig. 7f; Supplementary Fig. 8e), suggesting that IFNβ induction of Gal-9 expression may be affected by EGFR signaling in lung cancer cells. Unlike cancer cells, primary human macrophages constitutively express Gal-9 (Fig. 7g).

Galectin mRNAs lack signal peptide coding sequence, and their proteins are synthesized by cytosolic ribosomes and do not enter the classical ER–Golgi secretory pathway[21,50]. This raises an important question of how galectins leave the cell to interact with its potential glycan ligands, which are located mostly on the cell surface or in the extracellular matrix. It was recently reported that Gal-9 and TIM-3 are secreted by exocytosis as a complex in acute myeloid leukemia cells that co-express these two molecules[51] although how and where the Gal-9/TIM-3 complexes are formed is not clear. Interestingly, we found that IFNβ facilitated Gal-9 secretion from tumor and myeloid cells, and its secretion was further augmented by the presence of IFNγ (Fig. 7h–k). Interestingly, although IFNs failed to strongly upregulate cellular Gal-9 in the EGFR mutant cell lines (H1650, H1975, HCC827), they efficiently induced its secretion (Fig. 7j). In addition, while Gal-9 is constitutively expressed in the monocytic cell line THP-1

(Fig. 7a) and primary macrophages (Fig. 7g), its secretion is still regulated by interferons (Fig. 7i, k). These results suggest that IFNs independently regulate Gal-9 expression and secretion, and increased secretion of Gal-9 in the presence of interferons is not merely an indirect effect of upregulated Gal-9 expression. Analysis of data from multiple cancer types in The Cancer Genome Atlas (TCGA) and the Cancer Cell Line Encyclopedia (CCLE) databases revealed co-expression of Gal-9 with interferon-stimulated genes (ISGs; Fig. 7l, Supplementary Fig. 9), suggesting that Gal-9 is similarly regulated by interferon signaling in human cancers. Together, these results suggest that IFNs independently upregulate Gal-9 expression and secretion in both immune cells and cancer cells.

To further characterize Gal-9 expression in the settings of tumor immune response, we reanalyzed publicly available single-cell RNA-seq data of human melanoma[10]. Consistent with our data, cells that express high Gal-9 levels are mostly antigen-presenting cells (APCs), including B cells, dendritic cells, and macrophages (Fig. 8a–c). Interestingly, Gal-9 is also highly expressed in $T_{reg}$ cells (Fig. 8a–c). TILs from non-responders to anti-PD-1 therapy express much higher levels of Gal-9 compared to responders (Fig. 8d), suggesting that in certain human cancers combined blockade of PD-1 and Gal-9 could be an effective treatment strategy, as we have validated in animals (Supplementary Fig. 4e–h).

Overall, the data suggested a mechanism of Gal-9-mediated adaptive immune resistance in the TME. In growing tumors, IFNβ is produced by dendritic cells and cancer cells[52], while IFNγ is released by activated CD8 T cells. A concerted action of IFNβ and IFNγ upregulates Gal-9 expression in APCs (B cells, dendritic cells, and tumor-associated macrophages) and cancer cells and its secretion from these cells to dampen antitumor response by inducing T cell death (Fig. 8e).

## Discussion

The study provides evidence that PD-1 interacts with Gal-9 and TIM-3 to attenuate Gal-9/TIM-3-induced apoptosis of PD-1+TIM-3+ T cells in cancers, and demonstrates that Gal-9 is upregulated by the inflammatory cytokines IFNβ and γ, and targeting Gal-9 can be an effective strategy for cancer immunotherapy. Our findings shed new lights on the intricate war between cancer cells and the immune system, uncover a molecular mechanism of exhausted T cell persistence that involves the interactions between PD-1 and the Gal-9/TIM-3 cell death pathway, and demonstrate Gal-9 as a promising target for future generations of immune checkpoint therapy.

As its name implies, the *PDCD1* gene encoding the programmed cell death protein 1 (PD-1) was initially identified as one of the genes induced upon programed cell death[53], but subsequent studies have failed to show the activity of its protein product in inducing T cell apoptosis. By contrast, TIM-3, another

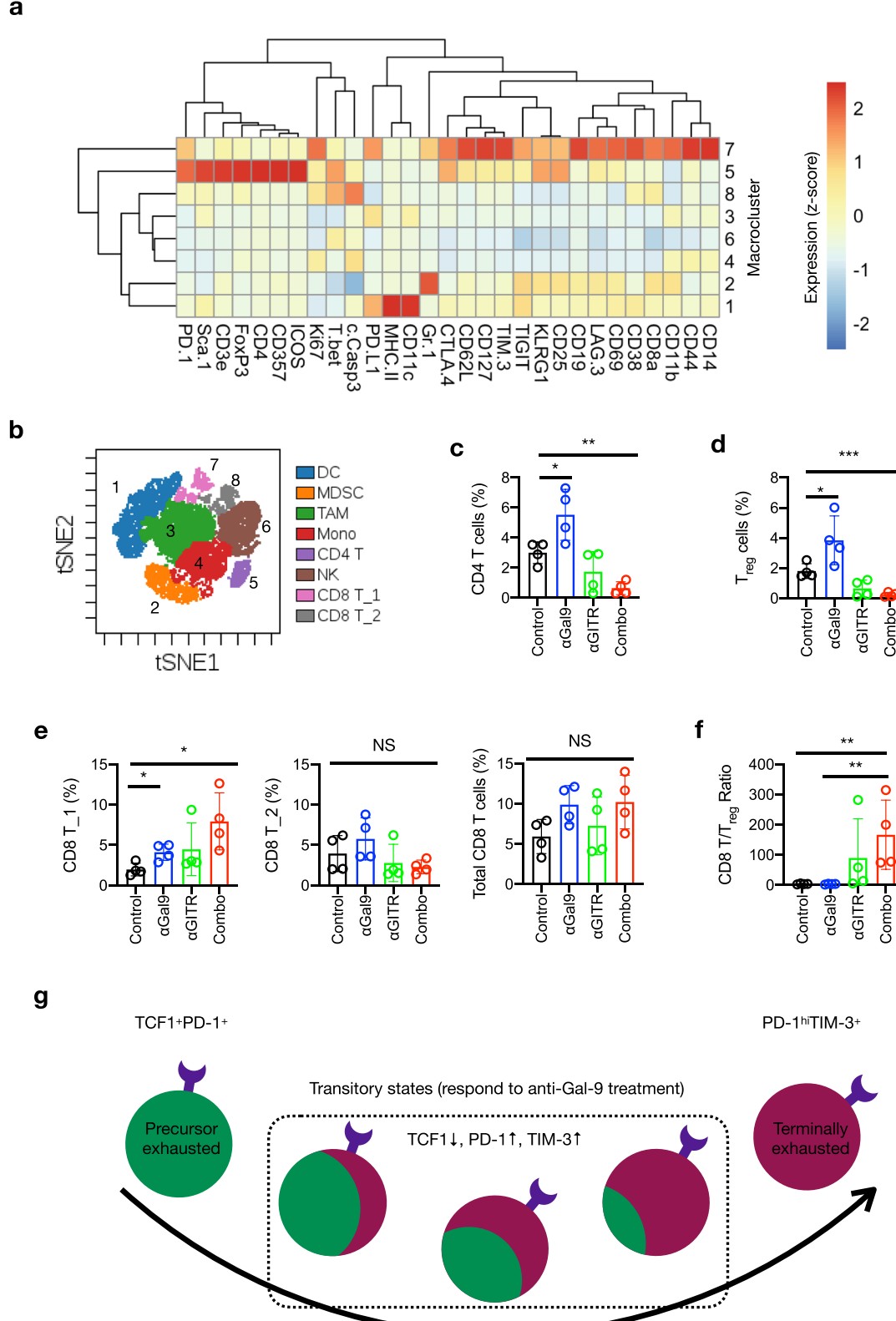

T cell inhibitory receptor co-expressed with PD-1 in exhausted T cells, mediates T cell apoptosis upon engagement by its ligand galectin-9[22]. Co-expression of PD-1 and TIM-3 marks exhausted CD8 T cells in both mouse and human tumors. By identifying the TIM-3 ligand Gal-9 as a PD-1 binding protein and providing evidence that PD-1 inhibits TIM-3+ T cell apoptosis through interacting with Gal-9, we unveiled a novel function for PD-1 that

contributes to the persistence of exhausted T cells in tumors. Our data suggest that homotypic crosslinking of TIM-3 by Gal-9 through its two CRDs results in (TIM-3/Gal-9/TIM-3)$_n$ lattices that amplify cell death signals to promote apoptosis; in PD-1+ TIM-3+ T cells, co-expressed PD-1 competes with TIM-3 to bind the C-CRD of Gal-9, facilitates the formation of (TIM-3/Gal-9/PD-1)$_n$ lattices at the expense of TIM-3/Gal-9/TIM-3)$_n$ lattices,

**Fig. 6 Anti-Gal-9 therapy targets specific tumor-infiltrating T cell populations. a** Heatmap showing differential marker expression in CD45$^+$ TIL clusters identified by analysis of CyTOF data using viSNE and FlowSOM. **b** Annotation of TIL populations based on differential marker expression as shown in (**a**) and Supplementary Fig. 6. DC, dendritic cell; MDSC, myeloid-derived suppressor cell; TAM, tumor-associated macrophage; Mono, monocytes; CD4 T, CD4 T cell; NK, natural killer cell; CD8 T_1, CD8 T cell subset 1; CD8 T_2, CD8 T cell subset 2. **c–e** T cell subset frequency in total CD45$^+$ TILs. **c** CD4 T cells. Control vs αGal-9, $P = 0.0337$; control vs combo, $P = 0.0013$. **d** T$_{reg}$ cells. Control vs αGal-9, $P = 0.0327$; control vs combo, $P = 0.0009$. **e** CD8 T cell subsets. CD8 T_1 subset: control vs αGal-9, $P = 0.0155$; control vs combo, $P = 0.0165$. **f** CD8 T/T$_{reg}$ ratios in TILs from indicated treatment groups. Control vs combo, $P = 0.0022$, αGal-9 vs combo, $P = 0.0021$. **g** Proposed model of Gal-9 inhibition-elicited T cell response in tumor. The proliferating transitory T cells in the process of precursor exhausted T cells differentiation into terminally exhausted T cells are the major responders to anti-Gal-9 treatment. Unpaired two-tailed $t$ tests were used for comparing means between treatment groups in (**c–f**). $n = 4$ mice in each treatment group. Error bars represent SD. Source data are provided as a Source data file.

and reduces TIM-3-mediated cell death. Attenuation of TIM-3$^+$ T cell death by PD-1 is consistent with a previous study showing that PD-1 is critical for CD8 T cell survival and the long term maintenance of exhausted T cell populations[54]. PD-1$^+$TIM-3$^+$ T cells are produced in tumors as TCF1$^+$PD-1$^+$TIM-3$^-$ precursor exhausted cells differentiate into PD-1$^{hi}$TIM-3$^+$ terminally exhausted T cells. These effector-like PD-1$^+$TIM-3$^+$ transitory T cells are essential for basal tumor control and provide the bulk of anti-tumor immunity in response to immunotherapy. The persistence of these cells in the TME may be attributed at least in part to their relative resistance to Gal-9-induced apoptosis.

Gal-9 and TIM-3 do not have an exclusive ligand-receptor relationship as they have multiple binding partners involved in different signaling pathways[55]. Although Gal-9-binding proteins are also found on some other immune cells, such as myeloid cells[56,57], our data show that Gal-9 inhibition in tumor-bearing mice selectively expands a subset of intratumoral TIM-3$^+$ CD8 T cells and CD4 T cells, including immunosuppressive T$_{reg}$ cells, whereas the frequencies of myeloid cells were largely unaffected. Anti-Gal-9 expanded proliferating transitory PD-1$^+$TIM-3$^+$ CD8 T cells that exhibit mixed activated/memory phenotypes, but the antitumor activity of these cells is likely suppressed by co-expanded TIM-3$^+$ T$_{reg}$ cells. Indeed, it has been shown that TIM-3$^+$ T$_{reg}$ cells are activated T$_{reg}$ cells that have superior immunosuppressive activity compared with their TIM-3$^-$ counterparts[44,45]. The combination of anti-Gal-9 and an agonist GITR antibody led to further expansion of the CD8 T cell subset and a near-complete depletion of T$_{reg}$ cells, producing synergistic antitumor effects. Such synergistic effects suggest that CD8 T cells and T$_{reg}$ cells rescued by anti-Gal-9 have exaggerated response to GITR stimulation. Why the agonistic GITR antibody DTA-1 expands CD8 T cells but depletes T$_{reg}$ cells is less clear, but it is known that T$_{reg}$ cells constitutively express higher levels of GITR compared to conventional T helper (Th) cells[58]. The T$_{reg}$ depletion function requires activating Fcγ receptors, suggesting the involvement of antibody-dependent cell-mediated cytotoxicity (ADCC) or phagocytosis (ADCP)[59]. High GITR levels on T$_{reg}$ cells may reach a certain threshold that triggers such processes, whereas lower levels of GITR on CD8 T cells mediate T cell co-stimulation. Combination of anti-Gal-9 with anti-PD-L1 appears to have lower antitumor therapeutic efficacy than the combination with anti-GITR, probably because PD-1 blockade, while recovers dysfunctional PD-1$^+$ CD8 T cells, also enhances PD-1$^+$ T$_{reg}$ cell-mediated immunosuppression[60–62]. In fact, PD-1$^+$ T$_{reg}$ cells amplified by PD-1 blockade have been shown to promote hyperprogression of cancer[63]. All in all these findings suggested that other therapeutic modalities that downregulate T$_{reg}$ cells and co-stimulate CD8 T cells are promising candidates for combination with Gal-9 inhibitors to treat cancer.

Similar to the PD-1 ligands PD-L1 and PD-L2[64], Gal-9 expression and secretion is upregulated by IFN signaling. Despite earlier reports of IFNγ-mediated regulation of Gal-9 expression in human endothelial and mesenchymal stromal cells[65,66], we found

that Gal-9 expression is primarily upregulated by IFNβ in tumor cells, whereas Gal-9 secretion is promoted by the combination of IFNβ and IFNγ. IFNβ is a key cytokine that mediates innate immune recognition of immunogenic tumors. Intratumoral dendritic cells (DCs) are the major producer of IFNβ as they sense danger signals from dying tumor cells[52], and activated T cells and natural killer cells are the primary source of IFNγ[67]. Thus, it is conceivable that during an anti-tumor immune response, immune cells and dying/dead tumor cells create a cytokine milieu in the TME that upregulates Gal-9 expression and secretion, and Gal-9 subsequently contributes to adaptive immune resistance by inducing T-cell apoptosis. Gal-9 is likely upregulated early on during cross-presentation of tumor antigens when intratumoral DCs sense danger signals from dying tumor cells, but is only secreted in large quantities after T cells are activated to produce IFNγ. Interestingly, T$_{reg}$ cells also express high levels of Gal-9, suggesting another mechanism for these cells to suppress immune response. Thus, our findings suggested that similar to IFN-mediated upregulation of PD-L1 expression[68], IFN-induced Gal-9 expression and secretion in tumors may be another mechanism of adaptive immune resistance that can be targeted for cancer immunotherapy.

Growing tumors produce IFNs as a result of activation of the cGAS-STING pathway, which is further augmented by cancer therapies[69,70]. Cancers of viral etiology are also associated with elevated IFNβ and Gal-9 expression[71]. Thus, rationally designed anti-Gal-9-based therapies in these settings is likely to improve patient prognosis. Against the backdrop of current cancer immunotherapies that aim to enhance T cell activity, our work demonstrates that inhibition of T cell death could be a viable strategy when combined with other therapeutic modalities, especially those that diminish T$_{reg}$ cells. We anticipate that our findings will open a new direction of cancer therapy to target a broad spectrum of malignancies, including those with primary or adaptive resistance to current immunotherapies.

## Methods

**Antibodies, cell lines and mice**. Antibodies used in this study for CyTOF and other assays with relevant information are listed in Supplementary Tables 1 and 2, respectively. The mouse colon adenocarcinoma cell line MC-38 was obtained from National Cancer Institute and maintained at 37 ℃, 5% CO$_2$, in DMEM media supplied with 10% fetal bovine serum (FBS). All other cell lines were obtained from ATCC. The human acute T cell leukemia cell line Jurkat (clone E6.1) and the mouse mammary carcinoma cell line EMT6 were maintained in RPMI-1640 supplemented with 10% FBS, and all other cell lines were maintained as recommended by ATCC. All cell lines were tested mycoplasma free. Four-week-old female C57BL/6J and BALB/cJ mice were purchased from The Jackson Laboratory and were allowed to acclimate to the housing facility for at least two weeks before experiments. This study complied with relevant ethical regulations for animal testing and research, and received ethical approval from The University of Texas MD Anderson Cancer Center (MDACC) Institutional Animal Care and Use Committee (IACUC). All animal experiments were performed in an MDACC AAALAC accredited barrier facility vivarium. Mice were maintained on a 12-h light/12-h dark cycle. Room temperatures are maintained within the range of 70 ℉ ± 2° F. The Humidity levels are maintained between 30–70%.

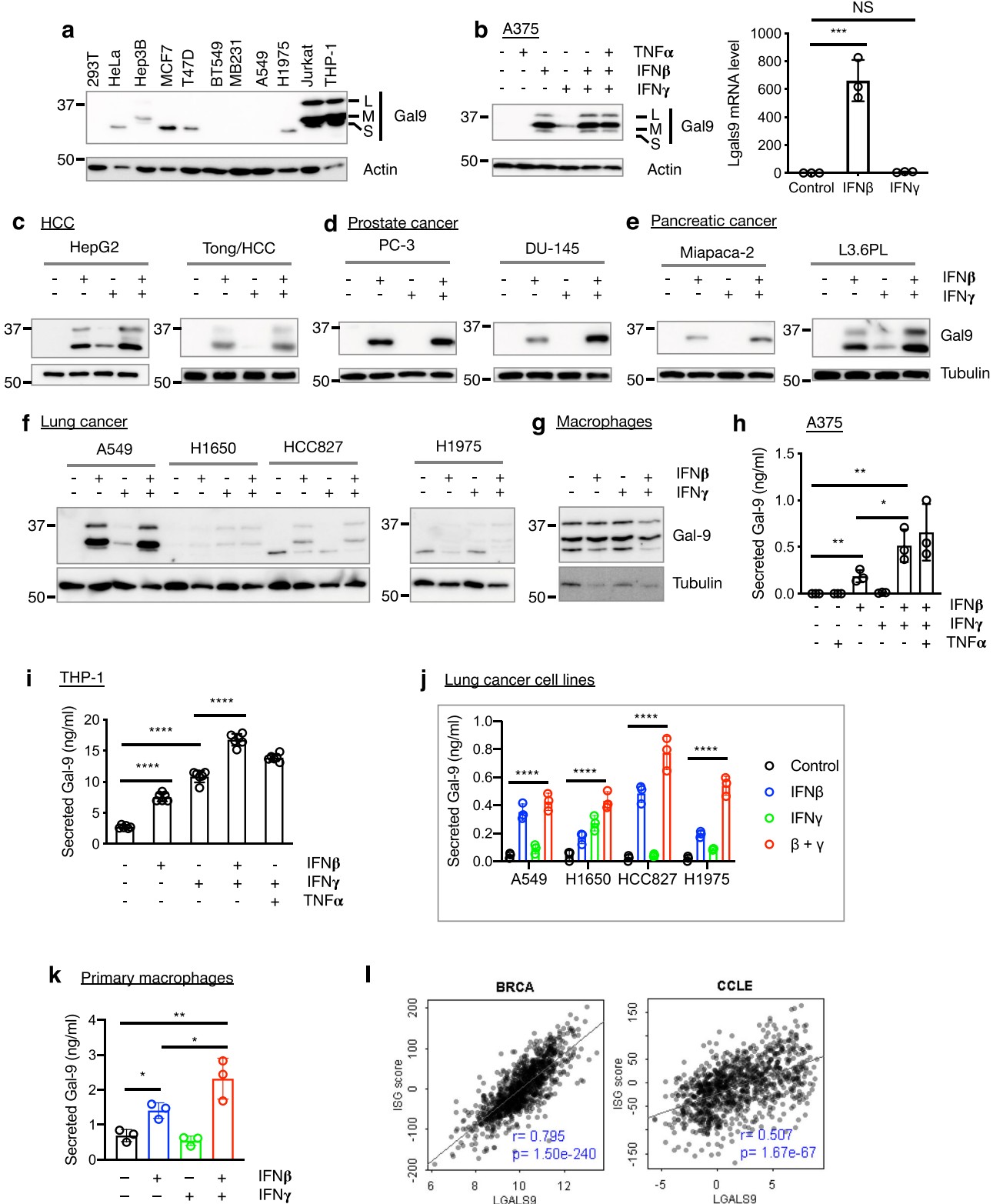

**Retro-lentiviral system for doxycycline-inducible gene expression**. The original retroviral vector pMA2641 that carries the gene for the Tet-On reverse transactivator Advanced (rtTA) driven by the viral LTR and the lentiviral vector pMA2867 with a doxycycline-inducible promoter have been described elsewhere[27]. The coding regions for EGFP and mCherry were deleted from those two vectors, giving rise to pMA2642 and pMA2868, respectively. Coding regions for proteins of interest were inserted into pMA2868 in the multiple cloning site downstream of the dox-inducible promoter. Site-directed mutagenesis was made using the Q5 Site-Directed Mutagenesis Kit (New England Biolabs). Another version of the pMA2868 vector (pMA2868H) was made by replacing the puromycin resistance

gene with the hygromycin-resistance gene. Coding region for human PD-1 and TIM-3 were amplified using the PicoMaxx High-Fidelity PCR System, cloned using the StrataClone PCR Cloning Kit (Agilent Technologies), sequenced, and finally cloned into pMA2868 or pMA2868H lentiviral vectors. In some versions of the constructs, the coding region for 3xFLAG tag was introduced 3' to the PD-1 or TIM-3 coding region by PCR (Supplementary Table 3). Retroviruses and lentiviruses were produced with the resulted viral constructs and the packaging plasmids pUMVC (for retroviruses) or psPAX2 (for lentiviruses), and the VSV-G envelope expressing plasmid pMD2.G, by co-transfection into 293T cells using polyethylenimine (PEI). Viruses were harvested and passed through 0.4 μm filter.

**Fig. 7 Interferon β and γ promote Gal-9 expression and secretion. a** Baseline expression of Gal-9 protein in cancer cell lines. L, M, and S denote the three common Gal-9 isoforms resulting from alternative pre-mRNA splicing. **b** Regulation of Gal-9 expression in A375 human melanoma cells by indicated inflammatory cytokines at protein (left) or mRNA (right) levels. Data are presented as mean values ± SD. Statistical differences were assessed using ordinary one-way ANOVA with Dunnett's multiple comparisons test. NS, not significant; ***$P < 0.001$. Control vs IFNβ, $P = 0.0002$; control vs IFNγ, $P = 0.9940$. **c–g** The effects of IFNβ and IFNγ on Gal-9 expression in cell lines of multiple cancer types and primary macrophages. **h–k** The effects of IFNβ and IFNγ on Gal-9 secretion from tumor cells and immune cells. **h** A375 melanoma cells. $n = 3$ independent experiments. Control vs IFNβ, $P = 0.0074$; IFNβ vs IFNβ + IFNγ, $P = 0.0388$; control vs IFNβ + IFNγ, $P = 0.0070$. **i** THP-1 monocytic leukemia cells. $n = 6$ independent experiments. Control vs IFNβ, $P < 0.0001$; control vs IFNγ, $P < 0.0001$; IFNβ vs IFNβ + IFNγ, $P < 0.0001$; IFNγ vs IFNβ + IFNγ, $P < 0.0001$. **j** Lung cancer cell lines. $n = 3$ independent experiments. Control vs IFNβ + IFNγ, $P < 0.0001$ for all the cell lines. **k** Primary macrophages. $n = 3$ independent experiments. Control vs IFNβ, $P = 0.0117$; IFNβ vs IFNβ + IFNγ, $P = 0.0395$; control vs IFNβ + IFNγ, $P = 0.0099$. **l** Expression correlation of *LGALS9* with ISGs in the TCGA BRCA dataset and all cancer cell lines in CCLE, as analyzed by linear regression (Pearson correlation with two-tailed *p*-values). Unpaired two-tailed *t* tests were used for comparing means between treatment groups in (**h–k**). Each circle represents one experiment. Error bars represent SD. Data shown in (**a–g**) are representative of three independent experiments.

Cells were transduced by incubation with viral supernatants in the presence of 6 µg/ml DEAE-dextran and centrifuged at 1200 × *g* at room temperature (RT) for 30 min. Selection with puromycin or hygromycin commenced 2 or 3 days after viral transduction and continued until all control cells died. Cells were incubated with 1 µg/ml doxycycline for one day before used in assays.

**CRISPR-Cas9 knockout of Gal-9 in Jurkat cells**. DNA oligos 5′-caccgGGCGA TGGTAGTATTCAAAC-3′ and 5′-aaacGTTTGAATACTACCATCGCCc-3′ were annealed and cloned into BsmBI-digested pLentiCRISPR v2 (Addgene) under the control of the type 3 RNA polymerase III promoter U6. Lentiviruses were generated, and Jurkat cells transduced and selected with puromycin as described above. Gal-9 knockout was confirmed by western blotting using a mouse anti-human Gal-9 antibody (1 µg/ml; clone OTI1G3, Bio-Rad).

**Gal-9 recombinant protein purification and coupling to sepharose beads**. The coding region of Gal-9 cDNA was cloned in-frame into pET21 (Novagen) between the NdeI and the XhoI sites. For protein expression, the resulting construct was transformed into E. coli strain BL21(DE3) (Sigma). Cells were allowed to grow in 2xYT medium to log phase (A600 = 0.6–0.7) and then IPTG was added to a final concentration of 0.1 mM. The culture was incubated at 20 °C overnight with constant shaking. Cells were then harvested by centrifugation and lysed in B-PER Reagent (Thermo) per the manufacturer's instruction. After centrifugation, Gal-9 was purified from the supernatant by affinity chromatography on a lactosyl-agarose column (Sigma). CNBr-activated Sepharose 4B (Sigma) was used to couple Gal-9 protein to Sepharose beads per the manufacturer's instructions.

To produce GST-fusion proteins of Gal-9 N- and C-CRDs, coding sequences for the N-CRD (aa 1–148) and C-CRD (aa 159–323) were amplified from human Gal-9 cDNA with PCR and cloned in-frame into the pGEX-6P-1 vector (GE Life Sciences) between the EcoRI and the XhoI sites. For GST-fusion protein expression, the resulting construct was transformed into E. coli strain BL21 (Sigma). Cells were allowed to grow in 2xYT medium to log phase (A600 = 0.7–0.8) and then IPTG was added to a final concentration of 0.1 mM. The culture was incubated at 25 °C overnight (for GST-9N) or 30 °C for 6 h (for GST-9C) with constant shaking. Cells were then harvested by centrifugation and lysed in B-PER Reagent (Thermo) per the manufacturer's instruction. After centrifugation, GST-fusion proteins were purified from the supernatant by affinity chromatography with glutathione-conjugated Sepharose beads (GenScript) per the manufacturer's instructions.

**Site-directed mutagenesis**. Constructs for expression of Gal-9 mutants were made with the NEB Q5 Site-Directed Mutagenesis Kit using the wildtype constructs as templates. Primers for this purpose were designed using the NEBaseChanger tool (http://nebasechanger.neb.com). Primer names and sequences for the Gal-9 CRD loss-of-function mutants (R65A and R239A) and the PD-1 glycosylation site mutants (N49Q, N58Q, N74Q, and N116Q) are listed in Supplementary Table 3.

**Identification of PD-1-binding proteins**. Jurkat cells engineered to express PD-1 with three C-terminal FLAG tags (PD-1.3F) with the above described doxinducible system were incubated with 1 µg/ml doxycycline and activated with 1 µg/ml anti-CD3 and 2 µg/ml anti-CD28 overnight. Cells ($3 \times 10^7$) were washed with PBS and lysed in a lysis buffer comprising 50 mM Tris HCl, 150 mM NaCl, 1% Triton X-100, pH 7.4. After centrifugation at 16,100 × *g* for 10 mins at 4 °C, supernatants were incubated with anti-FLAG M2 magnetic beads (Sigma) for 2 h followed by 3 washes in 5× TBST and one wash in 1× TBST. Bound proteins were then eluted in 30 µl SDS-sample buffer and subjected to protein identification by liquid chromatography with tandem mass spectrometry (LC-MS/MS).

**Immunoprecipitation/western blot analysis of protein interactions**. Jurkat cells that express 3×FLAG-tagged bait proteins were lysed in PTY buffer (50 mM HEPES, 50 mM NaCl, 5 mM EDTA, 1% Triton X-100, 50 mM NaF, 10 mM $Na_4P_2O_7$. pH to 7.4) and lysates were incubated for 2 h with anti-FLAG (M2) magnetic beads (Sigma). Beads were then wash 3 times with 5× TBS-T buffer and once with 1× TBS-T buffer. Bound proteins were eluted in SDS-sample buffer and subjected to western blot with specific antibodies. Mouse anti-Gal-9 antibody clone OTI1G3 (Bio-Rad), PD-1 (D4W2J) XP rabbit mAb (Cell Signaling Technology), and rabbit anti-TIM-3 antibody MAB23652 (R&D Systems) were used at 1:1000, 1:1000, and 2 µg/ml, respectively. Where indicated, cells were incubated with 2 µg/ ml Gal-9 in PBS at 4 °C for 1 h before lysis.

**Plate-based protein binding assays**. Unless otherwise indicated, Nunc Maxisorp ELISA Plates (BioLegend) were coated with recombinant Gal-9 at 2 µg/ml in PBS at 4 °C overnight. Wells were washed 3 times with wash/incubation buffer (PBS containing 0.05% Tween 20) and blocked for 1 h with 1% BSA in PBS. Wells were again washed and incubated with 2× serial dilutions of human IgG1.Fc, PD-1 ECD. Fc, TIM-3 ECD.Fc, or PD-L1 ECD.Fc, all from BPS Bioscience, starting at 2 µg/ml, in wash/incubation buffer for 1–2 h at room temperature. This was followed by incubation with 1:10,000 HRP-labeled goat anti-human IgG, Fc specific antibodies (Sigma). After final washes, TMB substrate solution (ThermoFisher Scientific) was added and the reaction was allowed to proceed for 20 min before terminated by the addition of $H_2SO_4$. Absorbances at 450 nm with a reference wavelength of 570 nm was then determined using a BioTek Synergy Neo2 plate reader.

To measure PD-1competition with TIM-3 for Gal-9 CRDs, plates were coated with 2 µg/ml GST-9N or GST-9C and incubated with increasing concentrations (0–2 µg/ml) of TIM-3 ECD.Fc in the presence of 0, 2, 5, or 10 µg/ml of PD-1 ECD. mFc (mouse IgG2a Fc; BPS Bioscience). Binding of TIM-3 ECD.Fc to the immobilized Gal-9 CRDs under these conditions were determined as described above. Data were fit to the built-in "Competitive inhibition" model of GraphPad Prism.

To measure Gal-9-mediated PD-1/TIM-3 ECD binding, plates were coated with 2 µg/ml histidine-tagged PD-1 ECD and incubated with 0.05 µg/ml TIM-3 ECD.Fc in the presence of increasing concentrations (0–1 µg/ml) of Gal-9. Binding of TIM-3 ECD.Fc to immobilized PD-1 ECD was determined as described above. Data were fit to the built-in "Allosteric sigmoidal" model of GraphPad Prism.

**DuoLink in situ proximity ligation assay**. We utilized the DuoLink in situ proximity ligation assay (PLA, Sigma-Aldrich) according to the manufacturer's protocol. In brief, Jurkat-Gal-9KO cells co-expressing PD-1 and 3×FLAG-tagged TIM-3 were treated with or without Gal-9 at 2 µg/ml for 6 h. Cells were cytospined onto coated microscope slides prior to being fixed and permeabilized with 4% paraformaldehyde and 0.1% Triton-100, respectively. Cells were immunolabeled with anti-FLAG M2 antibody (1:100, Sigma-Aldrich) and PD-1 (D4W2J) antibody (1:100, Cell Signaling Technology) overnight at 4 °C. After washing three times with Buffer A, samples were incubated with the secondary mouse PLUS and rabbit MINUS antibodies with attached PLA probes for 1.5 h at 37 °C in the dark. For ligation step, the samples were incubated for 30 min at 37 °C. Samples were finally counterstained with DAPI. Duolink data were processed and analyzed using ImageJ 2.0.0-rc-69/1.52p (NIH).

**Detection of insoluble Gal-9-glycoprotein lattice formation**. Jurkat T cells (1 × 10^6) overexpressing PD-1 or/and TIM-3 were treated with or without 2 µg/ml recombinant Gal-9 in cell culture medium for 2 h. Cells were then collected and lysed in 0.1 ml RIPA buffer for 15 mins at 4 °C. The cell lysates were centrifuged at 15,000 × *g* for 10 min at 4 °C. The supernatant was collected in new microtubes, and the pellet was lysed in SDS sample buffer. Proteins in the supernatant and pellet fractions were then analyzed by western blotting. Mouse anti-Gal-9 antibody clone OTI1G3 (Bio-Rad), PD-1 (D4W2J) XP rabbit mAb (Cell Signaling

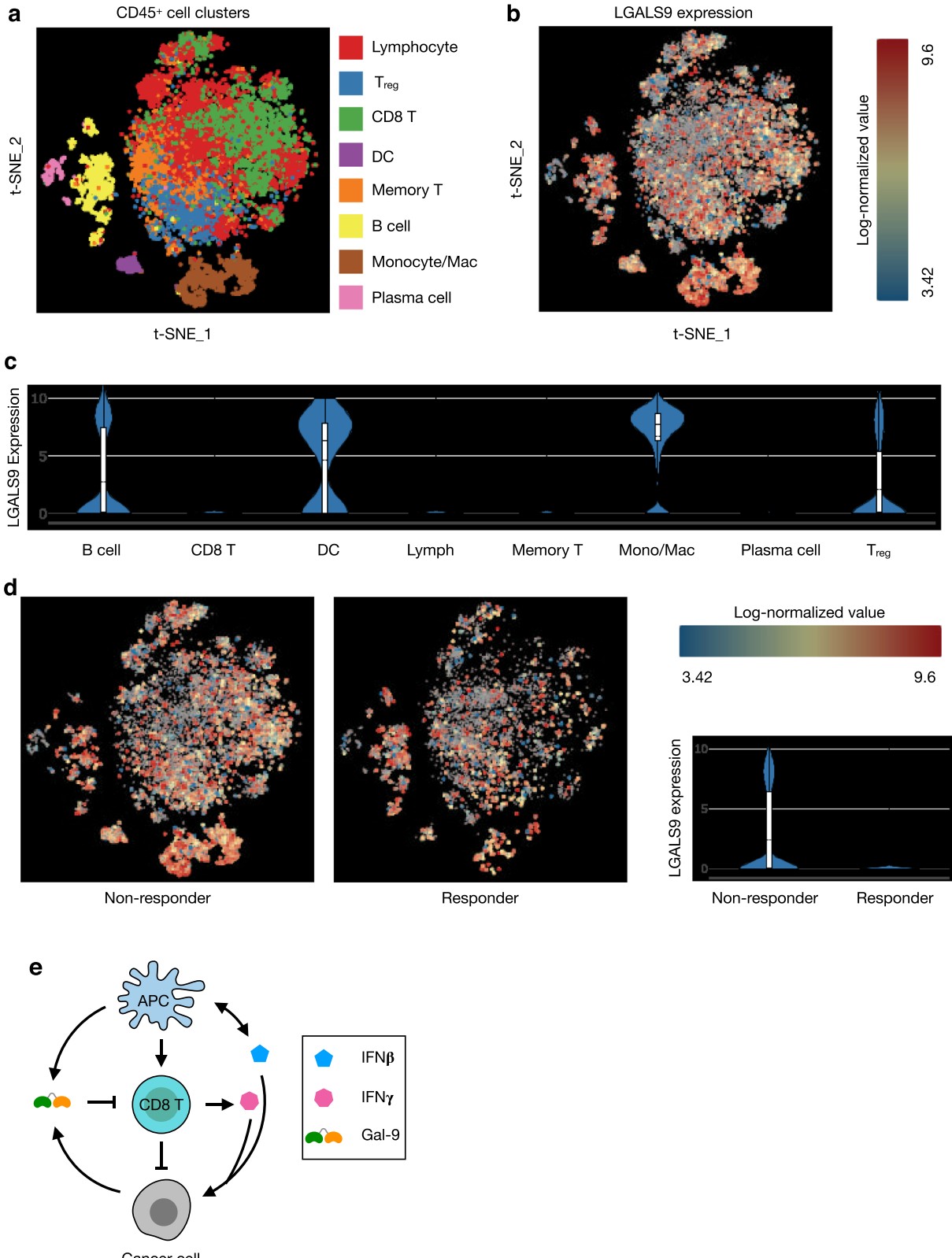

Technology), and rabbit anti-TIM-3 antibody MAB23652 (R&D Systems) were used at 1:1000, 1:1000, and 2 µg/ml, respectively.

**Isolation of human CD8 T cells**. Human peripheral blood mononuclear cells (PBMCs) were purchased from StemCell Technologies. CD8 T cells were isolated from human PBMCs by negative selection using the EasySep Human CD8+ T cell Isolation kit (StemCell Technologies) per the manufacturer's instructions. The CD8 T cell (CD3+CD8+) content of the isolated fraction is routinely >85%.

**Flow cytometric analysis**. PBMCs were incubated with Human TruStain FcX (BioLegend, 1:20) in staining/wash buffer (PBS containing 0.5% BSA and 0.02% NaN$_3$) for 10 min to saturate Fc receptors prior to staining with antibodies of interest. For antibody staining of cell surface proteins, cells were incubated with fluorescence-labeled antibodies (see Supplementary Table 2 for antibody dilutions) in 50 µl staining/wash buffer for 30 min in the dark. Cells were then washed and resuspended in 0.5 ml staining/wash buffer and subjected to flow cytometry. For staining intracellular Gal-9, cells were fixed in 3.7% formaldehyde for 20 min at

**Fig. 8 Reanalysis of single-cell RNA-seq data from melanoma TILs for Gal-9 expression.** Single-cell RNA-seq data of human melanoma TILs (GSE120575)[10] were reanalyzed with BBrowser2 (BioTuring). **a, b** t-SNE plots showing cell type composition (**a**) and Gal-9 expression (**b**) in melanoma TILs. **c** Violin plot showing expression of Gal-9 in melanoma TIL cell types. $n = 1455$ B cells, 305 plasma cells, 1391 macrophages/monocytes, 209 dendritic cells, 5459 lymphocytes, 3878 CD8 T cells, 1740 regulatory T cells, and 1773 memory T cells. **d** Gal-9 expression in non-responder ($n = 10,190$ cells) and responder ($n = 5110$ cells) to anti-PD-1 therapy. For violin plots in (**c, d**), whiskers indicate minima and maxima, lines inside box indicate medians and means, and bounds of box indicate 25th and 75th percentiles, respectively. **e** Interferon-induced Gal-9 expression and secretion as a potential mechanism of tumor adaptive immune resistance. In the TME, IFNβ (produced by APCs and tumor cells) and IFNγ (produced by activated CD8 T cells) induce Gal-9 expression and secretion by APCs and cancer cells. Gal-9, in turn, induces death of T cells and dampens the antitumor immune response.

room temperature, washed, and then permeabilized/stained in staining-permeabilization (SP) buffer (staining buffer containing 0.1% saponin) and fluorescence-labeled antibodies at the vendor-recommended dilution for 30 min. Cells were washed once in SP buffer and then resuspended in 0.5 ml staining buffer for analysis by flow cytometry. Data were acquired using a BD FACSCanto II flow cytometer with BD FACSDiva 8.0.1 software (BD Biosciences) and analyzed with FlowJo software (Becton Dickinson). In some experiments a microsphere was used as an internal counting standard to compare the cell density of a given population across treatment groups. In such cases, 10,000 BD FACS Accudrop Beads (BD Biosciences) per sample were added and a fixed number of 5000 bead events were acquired for each sample. In a forward scatter vs. side scatter plot, these beads were well separated from cells in the sample as a narrow band.

Mouse TILs were stained with the Zombie Violet Fixable Viability Kit (BioLegend) in PBS for 10 min, washed and incubate with Mouse BD Fc Block (BD Biosciences, 1:50) for 10 mins, and then stained for 30 min with fluorescence-labeled antibodies to mouse CD3, CD4, and CD8 (see Supplementary Table 2 for antibody dilutions). Cells were then washed in staining/wash buffer, and fixed/permeabilized with the eBioscience Foxp3/Transcription Factor Staining Buffer Set. Intracellular staining was performed with PE-labeled anti-mouse FoxP3 antibody (Invitrogen, 1:50). Frequency of the following cell types in total viable (Zombie Violet⁻) cells was determined: CD4 T cells (CD3⁺CD4⁺), CD8 T cells (CD3⁺CD8⁺) and T_reg cells (CD3⁺CD4⁺FoxP3⁺).

**Gal-9-induced cell death assay.** Cells ($3 \times 10^5$) in 1 ml culture medium were incubated with or without 2 μg/ml Gal-9. Two days later, cells were harvested and stained with APC-labeled anti-PD-1 and/or PE-labeled anti-TIM-3 antibodies (BioLegend, 1:50 each). After staining, cells were washed and resuspended in 0.5 ml staining/wash buffer containing 0.5 μg/ml 7-AAD (BioLegend). Counting beads (10000 BD Accudrop Beads) were added per sample and a fixed number of 3000 bead events were acquired for each sample. Data were analyzed with FlowJo as described above. Viable cells were gated according to normal scatter parameters (FSC/SSC) and 7-AAD exclusion, and cell survival was calculated using the following formula: Cell survival (%) = 100*(viable cell count in Gal-9-treated sample)/(viable cell count in control).

In some cases, cell death was determined by staining cells with FITC-labeled Annexin V (BioLegend, 1:20) and PI per the manufacturer's instructions.

**T cell-mediated cytotoxicity assay.** The coding region of the cDNA for the extracellular domain plus the transmembrane domain of human Fcγ receptor 2a (FcγR2a) was fused with that of the luciferase gene Luc2 and cloned into pMA2642. The retrovirus was produced in 293T cells and used to transduce A549 human lung cancer cells to express the FcγR2a-Luc2 fusion protein. CD8 T cells were isolated from human PBMCs using the EasySep Human CD8⁺ T Cell Isolation Kit (STEMCELL Technologies) and pre-activated with anti-CD3 and anti-CD28 antibodies. For cytotoxicity assay, target cells ($5 \times 10^3$ cells in 0.1 ml medium) were added to 96-well plate and cultured overnight. Medium was then replaced with 0.1 ml fresh medium containing 100 ng/ml mouse CD3 antibody OKT3, increasing concentrations of Gal-9, and $5 \times 10^4$ CD8 T cells pre-activated for 5 days with ImmunoCult Human CD3/CD28 T Cell Activator (STEMCELL Technologies). After incubation for 1 day, D-luciferin was added to a final concentration of 0.5 mg/ml and bioluminescence was measured using a BioTek Synergy Neo2 plate reader[72,73].

**Syngeneic mouse tumor models and treatments.** Mice were 6–8-week-old mice at the time of tumor inoculation. C57BL/6J mice were inoculated subcutaneously with $3 \times 10^5$ MC-38 mouse colon adenocarcinoma cells or $1 \times 10^5$ B16F10 mouse melanoma cells in phosphate-buffered saline (PBS). BALB/cJ mice were inoculated in the left mammary fat pad #4 with $1 \times 10^5$ EMT6 mouse mammary carcinoma cells in 50% PBS/50% Cultrex Basement Membrane Extract type 3 (R&D Systems). Mice were randomized into treatment groups 6–8 days later when tumors were palpable in most mice and treated with 100 μg rat IgG2b (isotype control), 100 μg anti-Gal-9 (RG9-1), 100 μg anti-GITR (DTA-1), or 100 μg anti-Gal-9 plus 100 μg anti-GITR (combo), all from Bio X Cell, via intraperitoneal injection. Treatment was repeated every three days for a total of three times. Tumors were measured with a digital caliper every 2 or 3 days, and tumor size calculated using the formula tumor volume (mm³) = L × W²/2, where L is the length and W is the width of the tumor. For survival curve analysis, mice were considered nonviable when tumors reached a volume of greater than 1000 mm³.

**Immunophenotyping of tumor-infiltrating leukocytes by CyTOF.** Surface and intracellular staining cocktail master mixes of metal-conjugated antibodies (Supplementary Table 1) were prepared by the MDACC Flow Cytometry and Cellular Imaging Core Facility prior to each experiment. Three days after the second antibody treatment, tumors were harvested from mice and single cells were obtained with the mouse Tumor Dissociation Kit and the gentleMACS Dissociator (both from Miltenyi Biotec) per the manufacturer's instructions. Cells were stained for viability with 25 μM cisplatin in RPMI-1640 for 1 min at room temperature and then washed. IgG Fc receptors were blocked by incubation with Mouse BD Fc Block (BD Biosciences, 1:50) in PBS containing 0.5% BSA and 0.02% NaN₃. Cells were then stained with a panel of heavy metal isotope-conjugated antibodies to 26 immune cell surface proteins (Supplementary Table 1). Samples were washed, fixed and permeabilized for 30 min with the eBioscience Foxp3/Transcription Factor Staining Buffer Set before being stained with heavy metal isotope-conjugated antibodies to FoxP3, T-bet, cleaved caspase 3, and Ki67 for 1 h. They were then washed again and incubated with 62.5 nM Cell-ID Intercalator-Ir (Fluidigm) in PBS containing 1.6% paraformaldehyde at 4 °C overnight. Samples were washed once in PBS, resuspended in water and acquired on a Helios mass cytometer (Fluidigm) at the MDACC Flow Cytometry and Cellular Imaging Core Facility. The acquired data were normalized using bead-based data normalization algorithm in the CyTOF software. Normalized data were manually gated in FlowJo to exclude normalization beads, debris, dead cells and doublets. DNA⁺Cisplatin⁻ singlets events were then exported to Cytobank[74] and subjected to viSNE clustering analysis on 28 marker channels with equal event sampling using 9986 events from each sample, for a total of 159776 events across all 16 samples. FlowSOM was then run on the coordinates of the viSNE map to generate FlowSOM metaclusters overlaid on the viSNE map. Heatmaps of marker intensities (generated using pheatmap R package) and channel-colored dot plots were used to assist in phenotype assignment to each metacluster. Where appropriate, the results of FlowSOM-driven metaclustering were further refined using the "automatic cluster gates" functionality and manual gating to create populations out of metaclusters. Statistics of population frequencies and marker intensities were exported to GraphPad Prism for graphing and statistical analysis.

**Real-time PCR.** Total RNA was isolated using Trizol reagent (Invitrogen). The first-strand cDNA was prepared using the PrimeScript 1st strand cDNA Synthesis Kit (Takara) according to the manufacture's protocol with 1 μg of total RNA. All RT-PCR reactions were performed in a 20-μl mixture containing 1× SYBR Green Master Mix (Takara), 0.2 μmol/L of each primer, and 2 μl of cDNA template. Primers for detection of human Gal-9 and glyceraldehyde 3-phophate dehydrogenase (GAPDH) cDNAs are listed in Supplementary Table 3. Real-time PCR was performed using the Applied Biosystem 7500 system under the following cycling conditions: 95 °C for 30 s, 40 cycles of 95 °C for 5 s, and 60 °C for 34 s, followed by the melting curve stage. The relative Gal-9 expression level was normalized to that of GAPDH.

**Differentiation of human monocyte-derived macrophages and assay for IFN-induced Gal-9 secretion.** Human PBMCs ($1\times10^6$ cells) in 1 ml RPMI 1640 media supplemented with 1% FCS were seeded in 24-well plates. After 2 h in culture, nonadherent cells were aspirated; and 1 ml fresh RPMI 1640 medium with 1% glutamax (Gibco) containing 50 ng/ml M-CSF (Sigma) and 1% FCS (v/v) was added and the medium was changed on day 2. Cells were stimulated on day 4 without or with IFNs (20 ng/ml) for 3 days. Conditioned medium was then harvested and measured for Gal-9 by ELISA using the Human Galectin-9 DuoSet ELISA kit (R&D Systems).

**In silico analysis of Gal-9 gene expression and correlation with patient survival.** Differential Gal-9 expression in cancer and normal tissue was analyzed with GEPIA[75]. Kaplan-Meier survival was estimated using KM-Potter (http://kmplot.com/analysis/). TCGA breast cancer gene expression data and clinical data were downloaded from The Cancer Genome Atlas (TCGA) data portal (https://tcga-data.nci.nih.gov/tcga/dataAccessMatrix.htm). CCLE RNAseq data were obtained from Expression Atlas (https://www.ebi.ac.uk/gxa/experiments/E-MTAB-2770/

Downloads). Interferon-stimulated gene (ISG) signature genes are from GSEA (http://software.broadinstitute.org/gsea/, HALLMARK_INTERFER-ON_ALPHA_RESPONSE genes). Gene set scores were calculated by summing up MAD modified Z scores of all signature genes.

**Statistical analysis**. Unless noted otherwise, data visualization and statistical analyses were performed using Prism 8.4.1 (GraphPad) or R. $P$ values < 0.05 were considered significant. Statistical tests are specified in figure legends.

**Reporting summary**. Further information on research design is available in the Nature Research Reporting Summary linked to this article.

## Data availability

Raw CyTOF data supporting the findings of this study have been deposited in FlowRepository [http://flowrepository.org/id/FR-FCM-Z2MJ] under the FlowRepository identifier FR-FCM-Z2MJ. Databases used in this study include The Cancer Genome Atlas (TCGA) (https://tcga-data.nci.nih.gov/tcga/), Expression Atlas (https://www.ebi.ac.uk/gxa/), the Molecular Signatures Database (https://www.gsea-msigdb.org/gsea/msigdb), and the National Center for Biotechnology Information Gene Expression Omnibus (NCBI-GEO) [https://www.ncbi.nlm.nih.gov/geo/query/acc.cgi?acc=GSE120575]. The remaining data are available within the Article, Supplementary Information or available from the authors upon request. Source data are provided with this paper.

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

## Acknowledgements

We thank Dr. Fu-Tong Liu for critically reading the manuscript, Dr. Jeng Cheng for editing and proofreading, and Dr. Zhenbo Han for technical assistance. This work was funded in part by the following: The University of Texas MD Anderson-China Medical University and Hospital Sister Institution Fund (to M.-C.H.); National Natural Science Foundation of China (81972186 and 31301160 to L.S.); Natural Science Foundation of Tianjin (20JCYBJC00360 and 16JCYBJC24400 to L.S.); China Scholarship Council (to L.S.); Conrad Biotech PTE.LTD.; Cancer Prevention & Research Institutes of Texas (RP160710); Breast Cancer Research Foundation; National Breast Cancer Foundation, Inc.; Center for Biological Pathways; The Odyssey Fellowship from The University of Texas MD Anderson Cancer Center (to C.-F.L.); Inha University Institution Fund (to J.-H.C.); and Inha University Research Grant (to J.-H.C.). Mass cytometry (CyTOF) was performed at the MDACC Flow Cytometry and Cellular Imaging Core Facility, which is funded in part by NCI Cancer Center Support Grant (CCSG) P30CA016672. Sequencing was performed at the MDACC Sequencing and Microarray Facility, which is funded by the grant P30CA016672 (SMF).

## Author contributions

R.Y., M.-C.H., and L.S. conceived the project; R.Y. and L.S. designed and performed the experiments, and analyzed data; C.-F.L., Y.-H.W., H.L., M.Y., J.-M.H., J.-H.C., C.-W.C., X.S., Y.D., and C.-K.C. performed experiments and analyzed data; J.Y. performed in silico gene expression analysis; R.Y., M.-C.H., and J.-L.H. wrote the manuscript; D.Y. provided scientific inputs; M.-C.H. and R.Y. supervised the entire project.

## Competing interests

The authors declare no competing interests.
