## [Peer Review File · Nature Communications]

REVIEWER COMMENTS

Reviewer #1 (Remarks to the Author): with expertise in cancer immunology

This manuscript by Hung and colleagues is a bit of a sprawling set of experiments documenting the biological activity of Galectin 9. There are broadly three largely non-overlapping stories housed in this manuscript, whose main points are: a) Gal-9 binds to PD-1 and this abrogates Gal-9-induced apoptosis (Fig 1-4), b) anti-Gal-9 enhances the responses of anti-GITR anti-tumor response and this is associated with an enhanced CD8/Treg ratio (Fig 5-6), c) Gal-9 expression is induced by interferons (beta and/or gamma) (Fig 7).

Major points:

To my knowledge, the data is largely novel, as galectin-9 had previously not been on my radar. Thus, it is a nice contribution to the literature. There is meticulous biochemical analyses performed to demonstrate galectin-9 binding to PD-1; however, the experiments to address the physiological relevance of this binding are not convincing (see below).

The finding that anti-Gal9 can have anti-tumor activity, particularly when combined with anti-GITR is the most interesting result presented. Unless I missed it, it is not clear how many times the critical experiments were replicated, but this would be needed to document the reproducibility of the report. The fact that the combination results in increased CD8/Treg ratio is an important finding. It would be important to understand why CD8 T increase and why Treg decrease. For both CD8 increase and Treg decrease, the following should be determined: a) absolute number, b) Ki67 should be documented to determine if anti-Gal9 +/- anti-GITR increases proliferation, c) bona fide cell death markers (PI, 7AAD, DAPI, or Caspase3/7, NOT annexin V) to determine if they are dying.

The expression pattern of galectin 9 is important. To enhance the reported conclusions, they should be repeated with two additional lung cancer cells lines and primary macrophages.

Specific Comments:

Fig S2c - Indicate how viability is determined

Fig S2d - Legend indicates "induction of cell death"; however, the y axis is likely not % cell death. I imagine this is cell survival as in S2f.

Fig S2e - How is apoptosis determined? Is it Annexin V+ and PI+? If Annexin V+ and PI- is included in the apoptosis panel, the author should note that many cells can reversibly express phosphatidylserine (Annexin V). Hence, if Annexin V+ PI- is counted as apoptotic, this should be confirmed with another assay such as PI, DAPI, 7AAD, Caspase 3/7.

Fig S2f - How is survival determined? Please indicate in legend.

Fig 4a - PD-1 and Tim-3 should be checked with flow cytometry to show that the same cell expresses both PD-1 and Tim-3.

Fig 4b - There are some reports that Gal-9 can activate T cells. As noted before, activated T cells can express phosphatidylserine. Thus, this figure should be confirmed with another readout of cell death.

Fig 4c-d - Since these are all relative percentages of four quadrants, it is not proper to plot a ratio of two of the quadrants only for quantification. The best way to do this experiment would be to use counting beads to count the T cells in every quadrant. If the authors wish to make a statement about the PD1- Tim3+ population, the absolute numbers should be quantified.

-I cannot follow the logic of these experiments. If making a statement about T cell death, it is unclear why a cell death readout (or even absolute numbers) is not utilized here.

-It is unclear what added information the experiment with PD-L1 beads provides. Even without PD1-PDL1 ligation, there is plenty of PD1 on the surface of the T cell. This should be sufficient to mediate binding to Gal-9.

-The most parsimonious explanation for this data is that Gal-9 can independently stimulate T cells (possibly not through Tim-3) and this can increase PD-1 and overcome PDL1 suppression. However, this does not necessarily support the author's conclusions.

Fig 4f - It is unclear why a correlation between PD-1/Tim-3 and Gal-9 would attest to their clinical relevance of the interactions in cancer. It is not as if the expression of one facilitates the mRNA expression of the other.

Fig 5b - This is a nice result. How many times was this experiment performed? For all experiments, this should be noted in the figure legend in order to assess reproducibility.

-Has the anti-Gal-9 antibody been tested previously in anti-tumor settings? This should be cited if so.

-What is the rationale for choosing GITR, which has not shown clinical efficacy in patients, as compared to anti-PD-1 and PD-L1 which is widely used in clinic?

-It would be of interest to repeat in combination with anti-PD-1 or PD-L1 blockade.

Fig 6 - For interpretability, it would be nice if the authors placed Fig S5d into main panel Fig 6A or vice versa.

-The cytof is fantastic for multiparametric analyses and some degree of surprising expression of surface markers may be expected. With that said, it is curious that CD8_1 (cluster 7 expresses Gr-1, there is almost unanimous expression of Ki67, and Sca-1, no population with CD19 staining (except maybe CD8_1, cluster 7). These raise the questions of specific staining on the cytof panel.

-How many times was the Cytof performed? If only once, the reduction in Treg, increase in CD8, and increase in CD8/Treg seen with anti-Gal9 + anti-GITR should be confirmed with flow cytometry.

-Fig S5D - Is the scale bar correct? In the heat map, there is no blue coloring.

-Fig 6J-K- These analyses are very difficult to interpret in the absence of isotype control staining and also a lack of absolute number counts. It is unclear if any of these differences are meaningful.

Minor points:

-Fig S2i - Please note that "survaval" is incorrectly spelled.

-For T cell mediated cytotoxicity assay, how long were CD8 T cells activated? What was their source? -Line 519 - "C56BL'6J" misspelled

-Line 298 0 "is produced by dendritic cells tumor cells".

Reviewer #2 (Remarks to the Author): with expertise in Tim-3 and T cell activation in cancer

Yang, Sun et al. report for the first time that galectin-9 (Gal-9) is a binding partner for PD-1, alongside its previously known binding to TIM3. First, by using immunoprecipitation and mass spectrometry they identified gal-9 as a candidate to bind PD1. By various biochemical assays they were able to give strong and compelling evidence of gal9/PD1 binding indeed takes place and is specifically mediated by its C-terminal carbohydrate recognition domain (CRD) and not by its N-terminal CRD, which is unlike TIM3 which can bind gal9 via both of its CRDs. They show that a gal9/PD1/TIM3 complex can indeed form, and provide evidence that PD1 can interfere with the apoptotic effect of gal9/TIM3 binding by competing with TIM3 over gal9 binding. Gal9 expression in human tumor predicts poor survival and prognosis in patients. In murine adenocarcinoma model, anti-gal9 combined with anti-GITR treatment attenuated tumor growth significantly, due to the combined effect of increased CD8 T cell anti-tumor immunity and depleted Tregs. Finally, tumor cells respond to interferon-beta within the TME by secreting gal9, which can induce T cell apoptosis .

The finding of PD1/Gal9 interactions and their influence on TIM3/gal9 interactions is a very novel concept presented by the authors. Why do the PD-1 and Tim-3 get enriched in the exhausted T cells and is there a common ligand that regulates their enrichment is an important question that has been raised before but not adequately addressed. The paper begins to address that issue. The authors have successfully shown that this interaction is possible in a series of well-designed biochemical in vitro experiments (Figure 1-3).

The authors have also provided evidence for the impact of PD1/gal9 interaction in vivo, using murine tumor model (figure 4-6), and also provided additional in vitro data showing how cancer cells can produce gal9 following interferon signaling (figure 7). However, there are some major concerns that need to be addressed.

Major points:

1. Figure 4c-e shows a shift activated T cells from a PD1-TIM3+ to a PD1+TIM3+ phenotype, as a readout for cell apoptosis. There are several problems with this analysis: while PD1+TIM3- and PD1+TIM3+ phenotypes of CD8+ T cells have been studied in much detail (for example in Singer, Wang et al, Cell 2016), PD1-TIM3+ cells are poorly described in the literature, and in tumor infiltrating CD8+ T cells they represent a very minor population in terms of numbers. Given this, the fact that so many T cells in Figure 4C are PD1-TIM3+ is puzzling, and the significance of the ratio shown PD1-TIM3+ and PD1+TIM3+, and the relation of this ratio to apoptosis mediated by gal9, is far from being understood. Equally puzzling is Figure 4D in which PDL1 beads are introduced, which do not seem to give any added value to the data. A much simpler measurements of T cell apoptosis in total TIM3+ and TIM3- cells can be performed using the simple and established readout of Annexin-V staining as in figure 4B should be done. The authors can also substantiate the data by measuring calcium flux, another process mediated by the gal9/TIM3 interaction (see Zhu et al NI 2005). Finally, the scheme in figure 4E is problematic- having only TIM3 and not PD1 induces death on the Jurkat cells in figure 4B, but this is not a physiological situation for tumor infiltrating cells which very rarely express only TIM3 and not PD1 together with TIM3.
2. The effect of addition of exogenous Gal9 does not appear to change much of binding/interaction between Tim-3 and PD-1 as observed in Figure 3b (compare lanes 4 and 8). Is the interaction observed in lane 4 due to endogenous Gal9 and can this interaction be inhibited by adding Galactose?
3. Anti-Gal9 blockade antibody alone increases both CD4 T cells and FoxP3+ Tregs. How does Gal9 blockade increase Tregs, it is simply because of Tim-3: gal9 blockade or Gal9 interaction with CD44 that has been shown to regulate iTreg development. Does Gal9: Tim-3 interaction also induce cell death in Tregs, in their hands. The effect anti-Gal9 antibody blockade of Treg expansion is overcome by a combination therapy of anti-gal9 and anti-GITR. There is no explanation why was GITR was chosen as a candidate for therapy in the first place. Also, did the authors not to try to combine gal9 with anti-TIM3 or PD1 antibodies?
4. The combined expression of both PD-1 and Tim-3 results in reduction in apoptosis of T cells mediated by Tim-3:Gal9 interaction, but it is not clear how would PD-1 interferes with the deletional process induced by Tim-3:Gal9 interaction.
5. Figure 6 raises several questions that need to be addressed. First, the CyTOF analysis according to the viSNE algorithm yield two supposedly distinct clusters of CD8 T cells (CD8 T_1 and CD8 T_2), yet the nature of their difference is not fully clear, except several markers and different PD1/TIM3 expression pattern. A much clearer distinction would have reached if certain accepted markers for effector-like (CX3CR1, SLAMF7) vs. naïve-like (TCF1, CXCR5, SLAMF6) would have been used. While the CD8 T_1 seem to be an effector population with some exhausted/dysfunctional markers, the CD8 T_2 population seem to be much less defined as it lacks CD44 and CD62L (therefore is not naïve-like) but also does not express exhaustion markers. Even more puzzling is the fact that although GITR is expressed on both CD8 T cell populations (supp Figure 6C), apparently the anti-GITR antibody does not significantly deplete it, unlike Tregs. Finally, in the text it is claimed that the combined anti-gal9 and anti-GITR depletes T regs more than the sole anti-GITR treatment, but in Figures 6C-D this difference is not statistically significant.

Minor points:

1. Abstract- the authors use the word "differentiation" (line 35) to describe the process of T cell exhaustion. I recommend use the phrase "T cell dysfunctional states" as the process is not a differentiation per se.
2. Figure 1D- The figure shows ELISA data of binding of plate-bound gal9 to PD1 or TIM3 Fc fusion proteins. An important point to note, at least for the sake of discussion, is that the binding of PD1

to gal9 is lower compared to TIM3. It is important for the idea of TIM3 and PD1 competing for gal9. This also suggest that a genuine competitive biochemical assay should be performed between PD1 and TIM3, to elucidate what might be the stoichiometry of binding is (how many PD1 molecules are needed to compete with one TIM3 molecule). This might begin to explain why does PD-1 interfere with Tim-3 mediated apoptosis.

3. Figure 3B- the authors show WB for PD1 and TIM3 following IP for TIM3. A reciprocal IP for PD1, followed by WB from PD1 and TIM3, will strengthen the authors message in this figure.

4. Figure 4A shows that co-expression of both TIM3 and PD1 in Jurkat cells can protect the cells against gal9 mediated apoptosis, compared to the sole expression of TIM3. In Figure 2D the authors show that introducing the N116Q mutation in a predicted glycosylation site disrupted its binding to Gal9. The message of Figure 2 will be very substantially strengthened if the authors will introduce PD1 with the N116Q mutation in Jurkat cells alongside TIM3 and gal9 and then measure apoptosis. The prediction would be that the N116Q mutant PD1 cannot protect the cells from the gal9 mediated apoptosis in the presence of TIM3.

5. In Figure 6E-F, Tregs seem to express more TIM3 following anti-gal9 treatment- what do they authors think this finding mean?

6. Line 298- the authors write that "In growing tumors, IFNbeta is produced by dendritic cells tumor cells"- clearly this sentence needs editing and proofreading.

In summary, the paper for the first time describes that Gal-9 might be an additional ligand for PD-1 and this might be one of the mechanisms by Gal9 binds to both Tim-3 and PD-1+ T cells on exhausted T cells and the exhausted T cells are spared from imminent deletion in the tumor microenvironment. The expression of Gal-9 in human tumors correlated with decreased survival in patients and blockade of Gal9 and GITR, synergistically enhance anti-tumor responses but the mechanisms by which these two antibodies promote anti-tumor immunity is not fully elucidated. The biochemical data convincingly shows that Gal9 indeed binds to both PD-1 and Tim-3.

Reviewer #3 (Remarks to the Author): with expertise in galectin and cancer

In the manuscript under consideration, Yang and colleagues studied two T-cell inhibitory receptors PD-1 and TIM-3 co-expressed during T cell exhaustion. The authors propose that PD-1 contributes to the persistence of PD-1+TIM-3+ T cells by binding to the TIM-3 ligand galectin-9 and attenuates Gal-9/TIM-3-induced cell death. The preclinical experiments performed suggest that Anti-Gal-9 therapy selectively expands intratumoral TIM-3+ cytotoxic CD8 T cells and immunosuppressive regulatory T cells while the combination of anti-Gal-9 and an agonistic antibody to the co-stimulatory receptor GITR induces synergistic antitumor activity by depleting Tregs. The authors propose that galectin-9 expression and secretion are promoted by interferon beta and gamma while bioinformatic analysis of TCGA database show that high Gal-9 mRNA levels correlates with poor prognosis in multiple human cancers. The authors conclude that their work uncovers a novel function for PD-1 and suggest that Gal-9 is a promising target for immunotherapy.

This is an interesting study. However, in my opinion, major work needs to be done before securing the manuscript publication. The conclusions drawn in this work are based on preliminary data.

Major comments

1) It is not clear whether this proposed mechanism is cancer-type specific or not. It is intriguing why the authors used a mouse model of colon cancer (MC38 cells) but didn't analyze the COAD-TCGA cohort in their in silico studies. The reviewer considers that this should be addressed and other mouse models of cancer should be analyzed.

2) Related to the previous point there are several Single cell RNA-Seq studies from patients responding and non-responding to anti PD-1 therapy. The data from these studies is publicly available and T cell types could be analyzed to test the hypothesis and validate the results. The reviewer would like to see the analysis of these datasets

3) The manuscript includes several models as panels of figures: 1a, 3a, 4e and 7g. Importantly, Figure 1 a is based on published literature. The reviewer considers that this should NOT be included in the manuscript.

4) Figure 2d has a quantification of the WB without error bars. How many times was this WB performed? The reviewer would like to see a triplicate, error bars and a statistical analysis associated to this.

5) The experiments performed in Jurkat cells should be validated at least in T cells obtained from PBMCs.

6) The in vivo experiment shows that the authors started treatment with tumors of a very small size. Can the authors confirm what was the tumor volume at day 7? How many times was the experiment repeated? The combo group has 4 tumor-bearing mice not responsive to the treatment (Fig 5b)., however this doesn't seem to be represented in the growth curve in Figure 5c where the error bars for the combo group are very small. Can the authors explain? What is the escape mechanism in those 4 tumors?

7) The authors made the conclusion "Interestingly, we found that IFN β facilitated Gal-9 secretion from tumor and myeloid cells, and its secretion was further augmented by the presence of IFN γ " based on Fig 7 e and f. This is based on the data obtained from 1 tumor cell line ONLY and 1 myeloid cell line ONLY. Further validation is crucial to draw such conclusion. The authors then go and make a model based on these conclusion in Figure 7g.

Point-by-Point Response Letter (NCOMMS-20-14491)

Reviewer #1 (Remarks to the Author): with expertise in cancer immunology

This manuscript by Hung and colleagues is a bit of a sprawling set of experiments documenting the biological activity of Galectin 9. There are broadly three largely non-overlapping stories housed in this manuscript, whose main points are: a) Gal-9 binds to PD-1 and this abrogates Gal-9-induced apoptosis (Fig 1-4), b) anti-Gal-9 enhances the responses of anti-GITR anti-tumor response and this is associated with an enhanced CD8/Treg ratio (Fig 5-6), c) Gal-9 expression is induced by interferons (beta and/or gamma) (Fig 7).

Major points:

Point #1: *To my knowledge, the data is largely novel, as galectin-9 had previously not been on my radar. Thus, it is a nice contribution to the literature. There is meticulous biochemical analyses performed to demonstrate galectin-9 binding to PD-1; however, the experiments to address the physiological relevance of this binding are not convincing (see below).*

The finding that anti-Gal9 can have anti-tumor activity, particularly when combined with anti-GITR is the most interesting result presented. Unless I missed it, it is not clear how many times the critical experiments were replicated, but this would be needed to document the reproducibility of the report. The fact that the combination results in increased CD8/Treg ratio is an important finding. It would be important to understand why CD8 T increase and why Treg decrease. For both CD8 increase and Treg decrease, the following should be determined: a) absolute number, b) Ki67 should be documented to determine if anti-Gal9 +/- anti-GITR increases proliferation, c) bona fide cell death markers (PI, 7AAD, DAPI, or Caspase3/7, NOT annexin V) to determine if they are dying.

Response: We thank the reviewer for the insightful comments. We have performed one new experiment to confirm the anti-tumor activity of anti-Gal-9/anti-PD-L1 combination in the MC-38 model, and the results are consistent with the initial findings (see Fig. below, Supplementary Fig.4a,b):

Supplementary Fig. 4a and b. New experiments that confirms the effects of anti-Gal-9 and combination therapies in MC-38 mouse tumor models. a, Average tumor growth. b, Log-rank (Mantel-Cox) tests for comparison of survival curves.

We apologize in case you missed it. The activity was also verified in the EMT6 triple negative breast cancer animal model in the original manuscript (now Fig. 5d-f in current version).

For CD8 increase and Treg decrease, the following have been determined:

- a)** We agree with the reviewer that the absolute number will be informative to evaluate the changes in cells and we have done that in in vitro assays with transduced Jurkat cells, and primary T cells from PBMCs (see Fig.4 below, and Fig. S7b,c).

Fig. 4. Co-expressed PD-1 protects TIM-3⁺ T cells from Gal-9-induced cell death. a-c, Jurkat cells transduced with indicated proteins individually or in combinations were treated with or without Gal-9 for two days and stained with PD-1/TIM-3 antibodies. Cell survival of relevant PD-1/TIM-3 subsets was

determined by flow cytometry with counting beads. **a**, Cells were gated based on FSC/SSC parameters and 7-AAD staining. **b** and **c**, Viable single cells equivalent to 3000 counting beads are shown in plots for each sample (10000 beads were added to each sample just prior to data acquisition). Numbers in plots indicate cell count in corresponding gates. **d** and **e**, Human CD8 T cells were incubated in ImmunoCult-XF T Cell Expansion Medium with or without Gal-9 in the presence of IL-2 and ImmunoCult Human CD3/CD28/CD2 T Cell Activator for 2 days and analyzed by flow cytometry with counting beads as described above for the survival of different PD-1/TIM-3 subsets. Numbers in plots indicate cell counts in corresponding quadrants.

For cells isolated from tumors, however, because of the multiple steps of sample handling during CyTOF sample preparation, including tissue digestion, filtration, centrifugation/washes, fixation/permeabilization, extracellular/intracellular staining, the recovery rates of viable cells vary considerably among samples. Smaller tumors usually suffer greater cell loss due to over-digestion and disproportional cell loss during the multiple steps of staining and washes (centrifugation/aspiration). In addition, as reported by others^{2,3} the cancer patient prognosis and efficacy of immune checkpoint therapy usually correlate with CD8 T cell/Treg ratio, rather than the absolute number of either cell type^{1,2}. Therefore, we follow the published literature by comparing the frequency of relevant immune cell subsets in total tumor-infiltrating CD45⁺ immune cells.

- b) Following the reviewer's suggestion, we have now documented Ki67, and indeed found highest Ki67 expression in the anti-Gal-9-expanded CD8 T₁ subset (see Figs. below, Fig. 6a and Supplementary Fig. 7i), supporting that proliferation contributes to its expansion.

Fig. 6a

Fig. 6a. Heatmap showing differential marker expression in CD45⁺ TIL clusters.

Fig. S7i. Density plot showing that the majority of CD8 T₁ (cluster 7) cells are TIM-3⁺Ki67⁺.

- c) We did measure cleaved (active) Caspase-3 expression by CyTOF and found that it mainly associated with Ki67 expression (Supplementary figure 7j). This probably reflects T cell activation rather than cell death, as it has been reported that c-Caspase-3 levels are associated with antigen-driven expansion of T cells and not with cell death³. Cell death in vivo may not be reliably detected as death cells are constantly removed by efferocytosis, and c-Casp3/7 are not unique markers for dying T cells as these caspases are also activated by T cell activation³. We have quantified cell death in vitro using assays that include dyes that measure membrane permeability (7-AAD, propidium iodide, and Zombie Violet), and counting absolute number of viable cells with counting beads (see Fig. S2a, Fig. 4a, and Fig. S7b).

Fig. S2a**Fig. 4a****Fig. S7b**
Fig. 2a. Gal-9 induced bona fide apoptosis in human T cells as shown by an increase in Annexin V+PI+ cells.

Fig. 4a. Gating and quantification of single, live human Jurkat cells using counting beads and the live/dead dye 7-AAD.

Fig. S7b. Gating and quantification of live mouse tumor-infiltrating T cell subsets by using counting beads and staining with indicated antibodies and the live/dead dye Zombie Violet.

Point #2: *The expression pattern of galectin 9 is important. To enhance the reported conclusions, they should be repeated with two additional lung cancer cells lines and primary macrophages.*

Response: Following the reviewer's comment, we have added three lung cancer cell lines (H1650, HCC827, and H1975 in Fig. 7f, j) and primary macrophages (Fig. 7g, k). We included two additional lung cancer cell lines in Fig. S8e. The results support our conclusion.

f Lung cancer**g Macrophages****j Lung cancer****k Macrophages**
Fig. 7f, g, j, and k showing the regulation of Gal-9 expression and secretion by IFNs in human lung cancer cell lines and primary macrophages.

Please see Fig. S8e for results from additional lung cancer cell lines.

Specific Comments:

Point #1: Fig S2c - Indicate how viability is determined

Response: Viable cells were gated according to normal scatter (FSC/SSC) parameters and PI exclusion, and quantified using counting beads. Viable cell number in the absence of Gal-9 was taken as 100%.

Point #2: Fig S2d - Legend indicates "induction of cell death"; however, the y axis is likely not % cell death. I imagine this is cell survival as in S2f.

Response: We apologize for the mistake and have corrected it in the figure legend in the revised manuscript.

Point #3: Fig S2e - How is apoptosis determined? Is it Annexin V+ and PI+? If Annexin V+ and PI- is included in the apoptosis panel, the author should note that many cells can reversibly express phosphatidylserine (Annexin V). Hence, if Annexin V+ PI- is counted as apoptotic, this should be confirmed with another assay such as PI, DAPI, 7AAD, Caspase 3/7.

Response: We agree with the reviewer and we have used additional parameters including PI/7AAD/Zombie Violet to identify apoptotic cells. The original Fig S2e showed Annexin V+PI-. We have changed it to show Annexin V+PI+ cells instead (Fig S2e).

e

Fig S2e. e, Dose-dependent induction of CD8 T cell apoptosis by Gal9 or GST-fusion proteins of its individual CRDs, as measured by annexin V-binding/PI staining assay.

Point #4: Fig S2f - How is survival determined? Please indicate in legend.

Response: Cell survival was determined by MTS assay. OD values in the absence of Gal-9 was taken as 100%. This is now indicated in legend: “f. Differential sensitivity of various cell types to Gal-9-induced cell death. Cell survival was determined by MTS assay. OD values in the absence of Gal-9 was taken as 100%”.

Point #5: Fig 4a - PD-1 and Tim-3 should be checked with flow cytometry to show that the same cell expresses both PD-1 and Tim-3.

Response: We agree with the reviewer that it is important to show that the same cell expresses both PD-1 and TIM-3. We have checked PD-1 and TIM-3 expression by flow cytometry and the data are now presented in Fig 4a-b.

Fig. 4a-b, Jurkat cells transduced with indicated proteins individually or in combinations were treated with or without Gal-9 and stained with PD-1/TIM-3 antibodies. Single viable cells were gated (**a**) and cell survival of relevant PD-1/TIM-3 subsets was determined by flow cytometry with counting beads (**b**).

Point #6: Fig 4b - There are some reports that Gal-9 can activate T cells. As noted before, activated T cells can express phosphatidylserine. Thus, this figure should be confirmed with another readout of cell death.

Response: We thank the reviewer for the helpful suggestions. We now include 7-AAD staining as a readout of cell death, and determine viability by quantifying viable cells (7-AAD⁻ events with normal FSC/SSC parameters) in relevant subsets using counting beads in flow cytometry (Fig. 4a,b).

Fig. 4a-b, Jurkat cells transduced with indicated proteins individually or in combinations were treated with or without Gal-9 and stained with PD-1/TIM-3 antibodies. Single viable cells were gated (**a**) and cell survival of relevant PD-1/TIM-3 subsets was determined by flow cytometry with counting beads (**b**).

Point #7: Fig 4c-d - Since these are all relative percentages of four quadrants, it is not proper to plot a ratio of two of the quadrants only for quantification. The best way to do this experiment would be to use counting beads to count the T cells in every quadrant. If the authors wish to make a statement about the PD1- Tim3⁺ population, the absolute numbers should be quantified. -I cannot follow the logic of these experiments. If making a statement about T cell death, it is unclear why a cell death readout (or even absolute numbers) is not utilized here.

Response: We have replaced the data with changes in absolute number of viable cells as quantified using counting beads. 7-AAD staining was used as a cell death readout to gate out dead cells (Fig. 4). Please see below for the new figure and its legend.

Fig. 4d and e, Human CD8 T cells were incubated in ImmunoCult-XF T Cell Expansion Medium with or without Gal-9 in the presence of IL-2 and ImmunoCult Human CD3/CD28/CD2 T Cell Activator for 2 days and analyzed by flow cytometry with counting beads after staining with PD-1 and TIM-3 antibodies. Viable cells were gated based on FSC/SSC parameters and 7-AAD exclusion.

Point #8: -It is unclear what added information the experiment with PD-L1 beads provides. Even without PD1-PDL1 ligation, there is plenty of PD1 on the surface of the T cell. This should be sufficient to mediate binding to Gal-9.

Response: We fully agree with the reviewer that the experiment with PD-L1 beads is unnecessary for our story and we have deleted it in the revised version.

Point #9: -The most parsimonious explanation for this data is that Gal-9 can independently stimulate T cells (possibly not through Tim-3) and this can increase PD-1 and overcome PDL1 suppression. However, this does not necessarily support the author's conclusions.

Response: Yes. There are indeed reports that Gal-9 can stimulates T cells. However, our data show that at 2 $\mu\text{g/ml}$ (the concentration used in this experiment) the predominant effect of Gal-9 is the cell death induction (Fig S2c). In addition, this assay was performed in the presence of strong T cell activators (anti-CD3, CD28, and CD2). Gal-9's contribution to T cell activation (and hence PD-1 upregulation) should be relatively small compared with these strong activators. Indeed, we now have data to prove that is the case (Fig. 4f). Please see below for the figure.

Fig. 4f. Flow cytometric analysis of PD-1 expression in CD8 T cells under indicated treatment conditions. Numbers denote percentage of PD-1⁺ cells.

Point #10: *Fig 4f - It is unclear why a correlation between PD-1/Tim-3 and Gal-9 would attest to their clinical relevance of the interactions in cancer. It is not as if the expression of one facilitates the mRNA expression of the other.*

Response: We agree with the reviewer that the correlation analysis of PD-1/Tim-3 and Gal-9 is irrelevant to our story and we have deleted it in the revised version.

Point #11: *Fig 5b - This is a nice result. How many times was this experiment performed? For all experiments, this should be noted in the figure legend in order to assess reproducibility. -Has the anti-Gal-9 antibody been tested previously in anti-tumor settings? This should be cited if so.*

Response: We thank the reviewer for the positive feedback. We performed this experiment twice with similar results (Fig. 5a-c and Supplementary Fig. 4a,b) and included the information in the figure legends. To our best knowledge the anti-Gal-9 antibody has not been tested in anti-tumor settings.

Point #12: *-What is the rationale for choosing GITR, which has not shown clinical efficacy in patients, as compared to anti-PD-1 and PD-L1 which is widely used in clinic?*

Response: In our initial experiments anti-Gal-9 monotherapy did not show long-lasting antitumor effects, and we reasoned that this could be due to insufficient T cell co-stimulation after inhibition of Gal-9, as Gal-9 had been shown to be required for efficient signaling of 4-1BB, a T cell co-stimulatory receptor of the TNFR superfamily. We therefore chose GITR, another co-stimulatory receptor of the same TNFR superfamily, in hope that GITR agonism could compensate for such reduction in co-stimulation. We later found that depletion of Tregs by anti-GITR is likely important for the therapeutic efficacy of the combination therapy. GITR agonism has been shown to generate effective antitumor immune response in animal models⁴. Results from a phase 1 trial (NCT01239134) of the GITR agonistic antibody TRX518 showed that TRX518 reduces both circulating and intratumoral Treg cells, although substantial clinical responses were not seen⁴. Another phase 1 trial investigating the combination of this GITR agonist with PD-1 blockade in patients with advanced solid tumors is ongoing (NCT02628574).

Point #13: *-It would be of interest to repeat in combination with anti-PD-1 or PD-L1 blockade.*

Response: Following the suggestion, we have tested combination with anti-PD-L1, which appears to be less efficient than combination with anti-GITR (Supplementary Fig. 4e-h). This is probably because analogous to Gal-9 inhibition that expands both CD8 T cells and Tregs, PD-1 blockade, while recovers dysfunctional PD-1+ CD8 T cells, also enhances PD-1+ Treg cell-mediated immunosuppression⁵⁻⁷. In fact, PD-1+ Tregs amplified by PD-1 blockade have been shown to promote hyperprogression of cancer⁸.

Supplementary Fig. 4e-h. The EMT-6 mouse breast cancer model was used to evaluate the efficacy of combined anti-Gal-9/anti-PD-L1 blockade. **e**, Treatment schedule. **f**, Tumor growth curves of individual mice bearing tumors inoculated at day 0 and subjected to indicated treatment. **g**, The average tumor growth of mice inoculated with tumor cells and subjected to the indicated treatments. Error bars represent s.e. of the means. **h**, Log-rank (Mantel-Cox) tests for comparison of survival curves.

Point #14: Fig 6 - For interpretability, it would be nice if the authors placed Fig S5d into main panel Fig 6A or vice versa.

Response: We sincerely thank the reviewer for the suggestion. We have placed the heatmap as Fig. 6a in the revised manuscript.

Point #15:- The cytof is fantastic for multiparametric analyses and some degree of surprising expression of surface markers may be expected. With that said, it is curious that CD8_1 (cluster 7 expresses Gr-1, there is almost unanimous expression of Ki67, and Sca-1, no population with CD19 staining (except maybe CD8_1, cluster 7). These raise the questions of specific staining on the cytof panel.

Response: Thank the reviewer for the input on CyTOF. Indeed, it is curious that Gr-1 expression in CD8 T₁. With that said, Gr-1+CD8+ T cells as a source of IFN γ has been reported in the literature^{9,10}.

For CyTOF staining, we used CyTOF-approved antibodies that have been extensively evaluated by the manufacturers and The MD Anderson Flow Cytometry and Cellular Imaging Core Facility, and followed well-established protocols for staining so we have full confidence in the quality of the data. Staining specificity is also attested by the high quality viSNE map that separates the expression of markers into distinct locations on the map. We have observed expected major immune cell populations that were distinctly identified in a well resolved viSNE map (Fig. 6a,b). Particularly, there is a clear phenotypic difference between the two CD8 T cell subsets. These all suggest that staining was specific for most of the antibodies in the panel.

CytoF profiles cells by comparing the relative staining intensity of cells for a panel of antibodies to selected set of cell markers. Different antibodies have different staining efficiency; It would be difficult to conclude anything based on the absolute staining intensity by different antibodies. Although it appears that many clusters stained positive for Ki67 and Sca-1 based on the raw staining intensity, the relative staining intensity is clearly different among the clusters after we scaled the median staining intensity of each marker across the clusters (We apologize that in original manuscript we used raw median staining intensity for the heatmap) (Fig. 6a). As expected, Ki67 and Sca-1 expression is highest in cluster 7 (CD8 T₁) and cluster 5 (CD4 T cells) (Fig. 6a).

Fig. 6a. Heatmap showing differential marker expression in CD45⁺ TIL clusters.

As for CD19 staining, the fact that no other CD19⁺ population was identified except some staining in cluster 7 may suggest that in this particular tumor model at such an early stage, the B cell population is too small to be detected, although we cannot completely rule out the possibility that the antibody did not stain well,

Point #16:-How many times was the Cytof performed? If only once, the reduction in Treg, increase in CD8, and increase in CD8/Treg seen with anti-Gal9 + anti-GITR should be confirmed with flow cytometry.

Response: Only once for this experiment. Following the comment, we have confirmed the results with flow cytometry (Fig. S7d-f).

Fig. S7d-f, Determination of T cell frequency and CD8 T cell/Treg ratio in MC-38 mouse tumors after indicated treatments by flow cytometry. d, Gating strategy. e, Frequency of CD4, CD8, and Tregs. f, CD8 T/Treg ratios in TILs from the 4 treatment groups.

Point #17:-Fig S5D - Is the scale bar correct? In the heat map, there is no blue coloring.

Response: Yes. The scale bar is correct, but the negative part is unnecessary because there were no negative staining intensity values. The original heatmap was generated by CytoBank using default settings with raw median staining intensity. We have redrawn the heatmap with the pheatmap R package and scaled the intensity for each marker across the clusters. Difference in marker expressions across the clusters are now much clearer (Fig. 6a).

Fig. 6a. Heatmap showing differential marker expression in CD45⁺ TIL clusters.

Point #18:-Fig 6J-K- These analyses are very difficult to interpret in the absence of isotype control staining and also a lack of absolute number counts. It is unclear if any of these differences are meaningful.

Response: We thank the reviewer's candid opinion. We have deleted these two panels in Figure 6.

Minor points:

Point #1:-Fig S2i - Please note that "survaival" is incorrectly spelled.

Response: We sincerely apologize for the typo. This has been corrected.

Point #2:-For T cell mediated cytotoxicity assay, how long were CD8 T cells activated? What was their source?

Response: CD8 T cells were isolated from human PBMCs using the EasySep Human CD8⁺ T Cell Isolation Kit (STEMCELL Technologies) and pre-activated for 5 days before used in the assay (see highlighted text in the Methods section).

Point #3:-Line 519 - "C56BL'6J" misspelled

Response: We apologize for the typo and it is corrected to "C57BL/6J" in the revised manuscript.

Point #4:-Line 298 0 "is produced by dendritic cells tumor cells".

Response: We sincerely thank the reviewer for the careful proof reading and have changed that to "dendritic cells and tumor cells"

Reviewer #2 (Remarks to the Author): with expertise in Tim-3 and T cell activation in cancer

Yang, Sun et al. report for the first time that galectin-9 (Gal-9) is a binding partner for PD-1, alongside its previously known binding to TIM3. First, by using immunoprecipitation and mass spectrometry they identified gal-9 as a candidate to bind PD1. By various biochemical assays they were able to give strong and compelling evidence of gal9/PD1 binding indeed takes place and is specifically mediated by its C-terminal carbohydrate recognition domain (CRD) and not by its N-terminal CRD, which is unlike TIM3 which can bind gal9 via both if its CRDs. They show that a gal9/PD1/TIM3 complex can indeed form, and provide evidence that PD1 can interfere with the apoptotic effect of gal9/TIM3 binding by competing with TIM3 over gal9 binding. Gal9 expression in human tumor predicts poor survival and prognosis in patients. In murine adenocarcinoma model, anti-gal9 combined with anti-GITR treatment attenuated tumor growth significantly, due to the combined effect of increased CD8 T cell anti-tumor immunity and depleted Tregs. Finally, tumor cells respond to interferon-beta within the TME by secreting gal9, which can induce T cell apoptosis .

The finding of PD1/Gal9 interactions and their influence on TIM3/gal9 interactions is a very novel concept presented by the authors. Why do the PD-1 and Tim-3 get enriched in the exhausted T cells and is there a common ligand that regulates their enrichment is an important question that has been raised before but not adequately addressed. The paper begins to address that issue. The authors have successfully shown that this interaction is possible in a series of well-designed biochemical in vitro experiments (Figure 1-3).

The authors have also provided evidence for the impact of PD1/gal9 interaction in vivo, using murine tumor model (figure 4-6), and also provided additional in vitro data showing how cancer cells can produce gal9 following interferon signaling (figure 7). However, there are some major concerns that need to be addressed.

Major points:

Point #1: *Figure 4c-e shows a shift activated T cells from a PD1-TIM3+ to a PD1+TIM3+ phenotype, as a readout for cell apoptosis. There are several problems with this analysis: while PD1+TIM3- and PD1+TIM3+ phenotypes of CD8+ T cells have been studied in much detail (for example in Singer, Wang et al, Cell 2016), PD1-TIM3+ cells are poorly described in the literature, and in tumor infiltrating CD8+ T cells they represent a very minor population in terms of numbers. Given this, the fact that so many T cells in Figure 4C are PD1-TIM3+ is puzzling, and the significance of the ratio shown PD1-TIM3+ and PD1+TIM3+, and the relation of this ratio to apoptosis mediated by gal9, is far from being understood. Equally puzzling is Figure 4D in which PDL1 beads are introduced, which do not seem to give any added value to the data. A much simpler measurements of T cell apoptosis in total TIM3+ and TIM3- cells can be performed using the simple and established readout of Annexin-V staining as in figure 4B should be done. The authors can also substantiate the data by measuring calcium flux, another process mediated by the gal9/TIM3 interaction (see Zhu et al NI 2005). Finally, the scheme in figure 4E is problematic- having only TIM3 and not PD1 induces death on the Jurkat*

cells in figure 4B, but this is not a physiological situation for tumor infiltrating cells which very rarely express only TIM3 and not PD1 together with TIM3.

Response: We thank the reviewer for the helpful comments. Indeed, PD1-TIM3+ cells represent a minor population of CD8 T cells in tumors. Our theory (and one of the major points we would like to make in this paper) is that this is because Gal-9 secreted by cells (including tumor cells, dendritic cells and monocytes/macrophages) in the TME preferentially induced apoptosis of the PD1-TIM3+ subpopulation. Unlike T cells in the TME, data in Fig. 4c (4d in the current version) are from in vitro assay with isolated and activated CD8 T cells. We postulated that under such conditions there are many PD-1-TIM-3+ T cells is because of activation-induced TIM-3 expression, and the lack of Gal-9 to induce apoptosis of these cells. Indeed, there was a disproportional decrease in the number of this subpopulation of cells after Gal-9 was added (Fig. 4d, e)).

Fig. 4d and e, Human CD8 T cells were incubated in ImmunoCult-XF T Cell Expansion Medium with or without Gal-9 in the presence of IL-2 and ImmunoCult Human CD3/CD28/CD2 T Cell Activator for 2 days and analyzed by flow cytometry with counting beads after staining with PD-1 and TIM-3 antibodies. Viable cells were gated based on FSC/SSC parameters and 7-AAD exclusion.

We agree that the original Figure 4D in which PDL1 beads are introduced did not add much to our study and we hence have deleted this panel.

In this revision, we have quantified different PD-1/TIM-3 subsets in control and Gal-9-treated samples by flow cytometry with counting beads, after staining cells with anti-PD-1 and anti-TIM-3 for phenotyping and with 7-AAD for cell death (see Fig. 4d above). We did not use simple Annexin V staining for this because these CD8 T cells consist of different PD-1/TIM-3 subsets that we want to compare. Thanks for the suggestion to use calcium flux as an index for apoptosis, however, there is evidence that it is not required for Gal-9-induced T cell apoptosis¹¹. And it is also not a commonly used readout for apoptosis, thus we did not measure it.

As suggested, we have deleted Fig. 4E to prevent confusion.

Point #2: *The effect of addition of exogenous Gal9 does not appear to change much of binding/interaction between Tim-3 and PD-1 as observed in Figure 3b (compare lanes 4 and 8). Is the interaction observed in lane 4 due to endogenous Gal9 and can this interaction be inhibited by adding Galactose?*

Response: We thank the reviewer for the insightful question. We have performed additional experiment and found that lactose did not reduces co-IP of PD-1/TIM3 (Fig. 3g). This and other data suggest that TIM-3 and PD-1 can interact with each other through their intracellular domains (ICDs), and Gal-9 crosslinks their extracellular domains (ECDs) to form galectin/glycoprotein lattices (Fig. 3).

Fig. 3. Characterization of TIM-3/Gal-9/PD-1 tri-molecular interaction. **a, b**, TIM-3 ECD binding to plate-immobilized GST-Gal-9C (**a**) or GST-Gal-9N (**b**) in the presence of increasing concentrations of PD-1

ECD. **c**, PD-1 ECD binding to plate-immobilized TIM-3 ECD or Gal-9. **d**, TIM-3 ECD binding to plate-immobilized PD-1 in the presence of increasing concentrations of Gal-9. **e**, Duolink assay of PD-1 and TIM-3 association in Gal-9 KO Jurkat cells co-expressing the two receptors with or without Gal-9. Dashed lines represent mean values; error bars represent s.d. Statistical differences were assessed using unpaired two-tailed t-tests. **f**, Jurkat cells expressing PD-1 (myc tagged) and TIM-3 (3xFlag tagged) individually or together were incubated with or without 2 $\mu\text{g/ml}$ exogenous Gal-9 followed by IP/Western blotting with indicated antibodies. **g, h**, IP/Western analysis of Jurkat cells expressing TIM-3 and 3xFlag tagged wildtype PD-1 or PD-1(N116Q) mutant, individually or in indicated combinations, in the presence or absence of lactose. **i-k**, Jurkat cells expressing PD-1 (**i**) or TIM-3 (**j**) or both (**k**) were incubated with or without Gal-9, and then lysed in a detergent buffer and centrifuged. Protein levels in the supernatants (S) and pellets (P) were determined by Western blotting with the indicated antibodies. **l**, Schematic diagram showing TIM-3/Gal-9/PD-1 tri-molecular interactions. TIM-3 and PD-1 dimerize through their intracellular domains. Gal-9 crosslinks TIM-3/PD-1 dimers with its N-CRD (green) and C-CRD (orange) to form galectin/glycoprotein lattices. Data (**a-i**) are representative of two independent experiments.

Point #3: *Anti-Gal9 blockade antibody alone increases both CD4 T cells and FoxP3+ Tregs. How does Gal9 blockade increase Tregs, it is simply because of Tim-3: gal9 blockade or Gal9 interaction with CD44 that has been shown to regulate iTreg development. Does Gal9: Tim-3 interaction also induce cell death in Tregs, in their hands. The effect anti-Gal9 antibody blockade of Treg expansion is overcome by a combination therapy of anti-gal9 and anti-GITR. There is no explanation why was GITR was chosen as a candidate for therapy in the first place. Also, did the authors not to try to combine gal9 with anti-TIM3 or PD1 antibodies?*

Response: Indeed it has been reported that Gal-9 enhances the stability and function of Tregs by interacting with CD44¹², yet we found that similar to conventional CD4 T cells, human Tregs are susceptible to Gal-9-induced cell death (Supplementary Fig. 7b, c). This suggests that in the TME, Gal-9's Treg killing activity dominates its Treg promoting activity. Tumor-infiltrating Tregs may be especially susceptible to Gal-9 killing as they express high levels of TIM-3 compared with their counterparts in the periphery^{13,14}, explaining why Gal9 blockade increases Tregs.

Fig. S7b, c. Like other T cells, human Tregs are highly sensitive to Gal9-induced cell death. **b.** Gating strategy. **c.** Survival of T cell subsets after Gal-9 treatment.

As stated in the Results section of the text (**Page 9**) and our response to Reviewer #1 Pt 12, below is the rationale to justify combination of anti-Gal-9/GITR:

In our initial experiments anti-Gal-9 monotherapy did not show long-lasting antitumor effects, and we reasoned that this could be due to insufficient T cell co-stimulation after inhibition of Gal-9, as Gal-9 had been shown to be required for efficient signaling of 4-1BB, a T cell co-stimulatory receptor of the TNFR superfamily. We therefore chose GITR, another co-stimulatory receptor of the same TNFR superfamily, in hope that GITR agonism could compensate for such reduction in co-stimulation. We later found that depletion of Tregs by anti-GITR is likely important for therapeutic efficacy of the combination therapy. GITR agonism has been shown to generate effective antitumor immune response in animal models⁴. Results from a phase 1 trial (NCT01239134) of the GITR agonistic antibody TRX518 showed that TRX518 reduces both circulating and intratumoral Treg cells, although substantial clinical responses were not seen⁴. Another phase 1 trial investigating the combination of this GITR agonist with PD-1 blockade in patients with advanced solid tumors is ongoing (NCT02628574).

We have tested combination with anti-PD-L1, which appears to have lower antitumor therapeutic efficacy than the combination with anti-GITR (Supplementary Fig. 4e-h). This is probably because PD-1 blockade, while recovers dysfunctional PD-1+ CD8 T cells, in the meantime also enhances PD-1+ Treg cell-mediated immunosuppression⁵⁻⁷. In fact, PD-1+ Tregs amplified by PD-1 blockade have been shown to promote hyperprogression of cancer⁸. Combination with TIM3 has not been attempted because the two presumably act on the same pathway and their combination most likely will not yield synergistic effect.

Supplementary Fig. 4e-h. The EMT-6 mouse breast cancer model was used to evaluate the efficacy of combined anti-Gal-9/anti-PD-L1 blockade. **e**, Treatment schedule. **f**, Tumor growth curves of individual mice bearing tumors inoculated at day 0 and subjected to indicated treatment. **g**, The average tumor growth of mice inoculated with tumor cells and subjected to the indicated treatments. Error bars represent s.e. of the means. **h**, Log-rank (Mantel-Cox) tests for comparison of survival curves.

Point #4: *The combined expression of both PD-1 and Tim-3 results in reduction in apoptosis of T cells mediated by Tim-3:Gal9 interaction, but it is not clear how would PD-1 interferes with the deletional process induced by Tim-3:Gal9 interaction.*

Response: We thank the reviewer for the comment. Our data suggests that PD-1 may work by competing with TIM-3 for binding to the C-CRD of Gal-9 (see Fig. 3 above and Fig. 4 below). So far little is known about the mechanism of Gal-9/TIM-3-mediated apoptosis in the literature. This will be an important direction to pursue in the future.

Fig. 4. Co-expressed PD-1 protects TIM-3⁺ T cells from Gal-9-induced cell death. a-c, Jurkat cells transduced with indicated proteins individually or in combinations were treated with or without Gal-9 for two days and stained with PD-1/TIM-3 antibodies. Cell survival of relevant PD-1/TIM-3 subsets was

determined by flow cytometry with counting beads. **a**, Cells were gated based on FSC/SSC parameters and 7-AAD staining. **b** and **c**, Viable single cells equivalent to 3000 counting beads are shown in plots for each sample (10000 beads were added to each sample just prior to data acquisition). Numbers in plots indicate cell count in corresponding gates. **d** and **e**, Human CD8 T cells were incubated in ImmunoCult-XF T Cell Expansion Medium with or without Gal-9 in the presence of IL-2 and ImmunoCult Human CD3/CD28/CD2 T Cell Activator for 2 days and analyzed by flow cytometry with counting beads as described above for the survival of different PD-1/TIM-3 subsets. Numbers in plots indicate cell counts in corresponding quadrants.

Point #5: *Figure 6 raises several questions that need to be addressed. First, the CyTOF analysis according to the viSNE algorithm yield two supposedly distinct clusters of CD8 T cells (CD8 T₁ and CD8 T₂), yet the nature of their difference is not fully clear, except several markers and different PD1/TIM3 expression pattern. A much clearer distinction would have reached if certain accepted markers for effector-like (CX3CR1, SLAMF7) vs. naïve-like (TCF1, CXCR5, SLAMF6) would have been used. While the CD8 T₁ seem to be an effector population with some exhausted/dysfunctional markers, the CD8 T₂ population seem to be much less defined as it lacks CD44 and CD62L (therefore is not naïve-like) but also does not express exhaustion markers. Even more puzzling is the fact that although GITR is expressed on both CD8 T cell populations (supp Figure 6C), apparently the anti-GITR antibody does not significantly deplete it, unlike Tregs. Finally, in the text it is claimed that the combined anti-gal9 and anti-GITR depletes Tregs more than the sole anti-GITR treatment, but in Figures 6C-D this difference is not statistically significant.*

Response: We sincerely thank the reviewer for the constructive comments. There is a clear phenotypic difference between the two subsets in Fig. 6a (compare clusters 7 and 8 for marker expression). We do not expect to see a distinct conventional T cell phenotype for these subsets because unlike conventional T cells in acute infection, CD8 T cells in tumors or chronic infection are dominated by a separate lineage of dysfunctional T cells differentiated from precursor exhausted T cells¹⁵⁻¹⁷. A mixed phenotype of CD8 T₁ (expression of activation/proliferation, memory, and exhaustion markers. Fig. 6, Fig. S5, S6, S7g-j) suggest that these are transitory T cells in the process of precursor exhausted T cells differentiation into terminally exhausted T cells. CD8 T₂ did not show changes among the treatment groups (figure 6e) and is not the focus of this study. In other tumor immunotherapies cells with phenotypes similar to those of CD8 T₁ provide the bulk of antitumor T cell response¹⁸⁻²⁴.

We aimed at identifying major immune cell populations that associate with treatment efficacy of anti-Gal9 therapy. As such we had to balance T cell markers with markers for other immune cells when we designed the antibody panel for CyTOF. Now that we have identified T cells as the major responding cells, we will further characterize these cells in future studies with more T cell-centric markers, including those suggested by the reviewer.

It is still not very clear why the agonistic GITR antibody DTA-1 expands CD8 T cells but depletes Treg cells. However, it is known that Treg cells constitutively express higher levels of GITR than conventional T cells²⁵. The Treg depletion function requires activating Fc γ receptors, suggesting the involvement of antibody-dependent cell-mediated cytotoxicity (ADCC) or phagocytosis (ADCP)²⁶. High GITR levels on Treg may reach a certain threshold that triggers such processes, whereas lower levels of GITR on CD8 T cells mediate T cell co-stimulation.

The exact statement in the text (p11) is-- “*In line with the findings previously reported²⁷, intratumoral T_{reg} cell frequency was significantly reduced in mice treated with the agonistic GITR antibody DTA-1 (Fig. 6d). Remarkably, anti-Gal-9 combined with anti-GITR led to a near-complete loss of T_{reg} cells (Fig. 6d)*”. We think this statement is consistent with our findings.

Minor points:

Point #1: *Abstract- the authors use the word “differentiation” (line 35) to describe the process of T cell exhaustion. I recommend use the phrase “T cell dysfunctional states” as the process is not a differentiation per se.*

Response: We thank the reviewer for the suggestion. We agree that phrase “T cell dysfunctional states” is appropriate and accurate to describe T cell exhaustion. We used the term “differentiation” because it is now generally accepted that T cell exhaustion is a differentiation state that is observed in the presence of persistent antigen and chronic T cell receptor (TCR) stimulation, when a stem-like precursor population ($PD-1^{+}TIM-3^{-}$) differentiate into terminally differentiated exhausted ($PD-1^{hi}TIM-3^{+}$) cells^{17,20,24}.

Point #2: *Figure 1D- The figure shows ELISA data of binding of plate-bound gal9 to PD1 or TIM3 Fc fusion proteins. An important point to note, at least for the sake of discussion, is that the binding of PD1 to gal9 is lower compared to TIM3. It is important for the idea of TIM3 and PD1 competing for gal9. This also suggest that a genuine competitive biochemical assay should be performed between PD1 and TIM3, to elucidate what might be the stoichiometry of binding is (how many PD1 molecules are needed to compete with one TIM3 molecule). This might begin to explain why does PD-1 interfere with Tim-3 mediated apoptosis.*

Response: We thank the reviewer for the excellent suggestion. We have performed experiments to found that PD-1 efficiently competed with TIM-3 for binding to Gal-9C ($K_i = 20.47$ nM) but not for binding to Gal-9N (Fig. 3a,b). Please see below.

Fig. 3a,b. PD-1 efficiently competes with TIM-3 for binding to the C-CRD but not to the N-CRD of Gal-9.

Point #3: Figure 3B- the authors show WB for PD1 and TIM3 following IP for TIM3. A reciprocal IP for PD1, followed by WB from PD1 and TIM3, will strengthen the authors message in this figure.

Response: We agree with the reviewer that a reciprocal IP is needed to strengthen our findings. We now have new data to show that reciprocal IP for PD-1 can pull down TIM-3. Please see Fig. 3g and h in revised manuscript. Please see below.

Fig. 3g, h. IP/Western analysis of Jurkat cells expressing wildtype PD-1, PD-1(N116Q) mutant, and TIM-3 individually or in indicated combinations, in the presence or absence of lactose.

Point #4: Figure 4A shows that co-expression of both TIM3 and PD1 in Jurkat cells can protect the cells against gal9 mediated apoptosis, compared to the sole expression of TIM3. In Figure 2D the authors show that introducing the N116Q mutation in a predicted glycosylation site disrupted its binding to Gal9. The message of Figure 2 will be very substantially strengthened if the authors will introduce PD1 with the N116Q mutation in Jurkat cells alongside TIM3 and gal9 and then measure apoptosis. The prediction would be that the N116Q mutant PD1 cannot protect the cells from the gal9 mediated apoptosis in the presence of TIM3.

Response: We thank the reviewer for the excellent suggestion. We have performed the experiment and demonstrated that PD-1 N116Q mutant failed to protect the cells from Gal-9-induced apoptosis mediated by TIM-3. Please see Fig. 4a-c below.

Fig. 4a-c, Jurkat cells transduced with indicated proteins individually or in combinations were treated with or without Gal-9 and stained with PD-1/TIM-3 antibodies. Single viable cells were gated (**a**) and cell survival of relevant PD-1/TIM-3 subsets was determined by flow cytometry with counting beads (**b, c**).

Point #5: *In Figure 6E-F, Tregs seem to express more TIM3 following anti-gal9 treatment- what do they authors think this finding mean?*

Response: One plausible explanation is that Treg cells with higher TIM-3 are more sensitive to Gal-9-induced cell death; and they are therefore preferentially rescued when Gal-9 is inhibited. TIM-3 expression in Treg is associated with increased suppression activity^{13,14,28}, and this may explain why anti-Gal-9 monotherapy does not result in a potent antitumor response.

Point #6: *Line 298- the authors write that “In growing tumors, IFNbeta is produced by dendritic cells tumor cells”- clearly this sentence needs editing and proofreading.*

Response: Thanks for the comments. We have corrected the sentence as “*In growing tumors, IFN β is produced by dendritic cells and cancer cells*²⁹”.

Reviewer #3 (Remarks to the Author): with expertise in galectin and cancer

In the manuscript under consideration, Yang and colleagues studied two T-cell inhibitory receptors PD-1 and TIM-3 co-expressed during T cell exhaustion. The authors propose that PD-1 contributes to the persistence of PD-1+TIM-3+ T cells by binding to the TIM-3 ligand galectin-9 and attenuates Gal-9/TIM-3-induced cell death. The preclinical experiments performed suggest that Anti-Gal-9 therapy selectively expands intratumoral TIM-3+ cytotoxic CD8 T cells and immunosuppressive regulatory T cells while the combination of anti-Gal-9 and an agonistic antibody to the co-stimulatory receptor GITR induces synergistic antitumor activity by depleting Tregs. The authors propose that galectin-9 expression and secretion are promoted by interferon beta and gamma while bioinformatic analysis of TCGA database show that high

Gal-9 mRNA levels correlates with poor prognosis in multiple human cancers. The authors conclude that their work uncovers a novel function for PD-1 and suggest that Gal-9 is a promising target for immunotherapy.

This is an interesting study. However, in my opinion, major work needs to be done before securing the manuscript publication. The conclusions drawn in this work are based on preliminary data.

Major comments

Point #1: *It is not clear whether this proposed mechanism is cancer-type specific or not. It is intriguing why the authors used a mouse model of colon cancer (MC38 cells) but didn't analyze the COAD-TCGA cohort in their in silico studies. The reviewer considers that this should be addressed and other mouse models of cancer should be analyzed.*

Response: Thank you for the comments. We have so far evaluated three syngeneic mouse tumor models (MC-38, EMT-6 and B16F10) and demonstrated that anti-Gal-9/GITR works in both the MC-38 and the EMT6 cancer models (Fig. 5). We also analyzed the COAD-TCGA cohort and found that Gal-9 expression did not correlate with patient survival (please see Fig.R1 below shown only for reviewer's evaluation).

Fig.R1. Correlation of Gal-9 expression with patient survival in the COAD-TCGA cohort.

Syngeneic animal models are very useful for testing the general efficacy of immunotherapy (immune checkpoint blockade in particular). However, these models do not usually predict efficacy in the same cancer types in humans. This is because the efficacy of immune checkpoint therapy largely depends on the immunogenicity of the tumor^{30,31}, and human cancer and animal models of the same cancer type often differ greatly in immunogenicity³². For example, PD-1/PD-L1 blockade works well for the MC38 mouse colon cancer model but not for human colon cancer patients, except for a small subset that is mismatch-repair-deficient and microsatellite-

instability-high (dMMR/MSI-H). Conversely, PD-1/PD-L1 blockade is effective for human melanoma but is not effective in the poorly immunogenic mouse B16F10 melanoma model^{33,34}.

Point #2: *Related to the previous point there are several Single cell RNA-Seq studies from patients responding and non-responding to anti PD-1 therapy. The data from these studies is publicly available and T cell types could be analyzed to test the hypothesis and validate the results. The reviewer would like to see the analysis of these datasets*

Response: We thank the reviewer for the suggestion. We have reanalyzed publicly available single-cell RNA-seq data of human melanoma²¹. Consistent with our data, cells that express high Gal-9 levels are mostly antigen presenting cells, including B cells, dendritic cells, and macrophages (Fig. 8a-c). Interestingly, Gal-9 is also highly expressed in Treg cells (Fig. 8a-c). Notably, TILs from non-responders to anti-PD-1 therapy express much higher levels of Gal-9 than responders (Fig. 8d), suggesting that in certain human cancers combination of PD-1/Gal-9 inhibition could be an effective treatment strategy, as we have validated in the EMT-6 mouse breast cancer model (Supplementary Fig. 4e-h).

Fig. 8

Fig. 8. Reanalysis of single-cell RNA-seq data from melanoma TILs for Gal-9 expression. Single-cell RNA-seq data of human melanoma TILs (GSE120575)²¹ were reanalyzed with BBrowser2 (BioTuring). **a** and **b**, t-SNE plots showing cell type composition (**a**) and Gal-9 expression (**b**) in melanoma TILs. **c**, Violin plot showing expression of Gal-9 in melanoma TIL cell types. **d**, Gal-9 expression in non-responder and responder to anti-PD-1 therapy.

Supplementary Fig. 4e-h. The EMT-6 mouse breast cancer model was used to evaluate the efficacy of combined anti-Gal-9/anti-PD-L1 blockade. **e**, Treatment schedule. **f**, Tumor growth curves of individual mice bearing tumors inoculated at day 0 and subjected to indicated treatment. **g**, The average tumor growth of mice inoculated with tumor cells and subjected to the indicated treatments. Error bars represent s.e. of the means. **h**, Log-rank (Mantel-Cox) tests for comparison of survival curves.

Point #3: *The manuscript includes several models as panels of figures: 1a, 3a, 4e and 7g. Importantly, Figure 1a is based on published literature. The reviewer considers that this should NOT be included in the manuscript.*

Response: We agree with reviewer's assessment, and we have deleted Fig 1a and others.

Point #4: *Figure 2d has a quantification of the WB without error bars. How many times was this WB performed? The reviewer would like to see a triplicate, error bars and a statistical analysis associated to this.*

Response: We apologize for our oversight. We performed this experiment 3 times. Please see a revised Fig. 2d with error bars included below.

Fig. 2d. Binding of Gal-9 to PD-1 is primarily mediated by the N116-linked glycan of PD-1. Lysates of Jurkat cells expressing 3xFLAG-tagged WT PD-1 or glycosylation site mutants were incubated anti-FLAG M2 magnetic beads. Bound proteins were eluted and subjected to Western blotting with PD-1 or Gal-9 antibodies.

Point #5: *The experiments performed in Jurkat cells should be validated at least in T cells obtained from PBMCs.*

Response: We agree with the reviewer that our results should be validated in primary T cells. Please see Fig. 4d and e below for results with primary T cells isolated from human PBMCs.

Fig. 4d and e, Human CD8 T cells were incubated in ImmunoCult-XF T Cell Expansion Medium with or without Gal-9 in the presence of IL-2 and ImmunoCult Human CD3/CD28/CD2 T Cell Activator for 2 days and analyzed by flow cytometry with counting beads after staining with PD-1 and TIM-3 antibodies. Viable cells were gated based on FSC/SSC parameters and 7-AAD exclusion.

Point #6: *The in vivo experiment shows that the authors started treatment with tumors of a very small size. Can the authors confirm what was the tumor volume at day 7? How many times was the experiment repeated? The combo group has 4 tumor-bearing mice not responsive to the treatment (Fig 5b)., however this doesn't seem to be represented in the growth curve in Figure 5c where the error bars for the combo group are very small. Can the authors explain? What is the escape mechanism in those 4 tumors?*

Response: We thank the reviewer for the questions. At day 7 the average tumor volume was 89.97 mm³. The experiment was repeated twice (Fig. 5a-c and Supplementary Fig 4a, b). Please note that all 8 mice in the combo group exhibited slower tumor growth before day 20, although tumors in 4 of the mice in this group eventually took off. At day 21 (time point cutoff for average tumor growth curve computation because beyond this time point many mice in control and monotherapy groups either died or had to be sacrificed due to excessive tumor burden) the 8 mice in the combo group have a mean tumor volume of 221.40 mm³ and a standard error (represented by error bar) of 75.44, which is consistent with what were shown in Fig. 5a-c.

The escape mechanism for anti-Gal-9/GITR therapy is currently unknown and will be the focus of our next project.

Point #7: *The authors made the conclusion “Interestingly, we found that IFN β facilitated Gal-9 secretion from tumor and myeloid cells, and its secretion was further augmented by the presence of IFN γ ” based on Fig 7 e and f. This is based on the data obtained from 1 tumor cell line ONLY and 1 myeloid cell line ONLY. Further validation is crucial to draw such conclusion. The authors then go and make a model based on these conclusion in Figure 7g.*

Response: We agree with reviewer that more tumor cell lines should be used to confirm our results. We have validated this with 3 more lung cell lines and primary macrophages (Fig. 7j, k). Please see below for the results.

Fig. 8j, k. Regulation of Gal-9 secretion by IFNs in human lung cancer cell lines and primary macrophages.

References

1. Jordanova, E. S. *et al.* Human leukocyte antigen class I, MHC class I chain-related molecule A, and CD8⁺/regulatory T-cell ratio: Which variable determines survival of cervical cancer patients? *Clin. Cancer Res.* **14**, 2028–2035 (2008).
2. Sato, E. *et al.* Intraepithelial CD8⁺ tumor-infiltrating lymphocytes and a high CD8⁺/regulatory T cell ratio are associated with favorable prognosis in ovarian cancer. *Proc. Natl. Acad. Sci.* **102**, 18538–18543 (2005).
3. McComb, S., Mulligan, R. & Sad, S. Caspase-3 Is Transiently Activated without Cell Death during Early Antigen Driven Expansion of CD8⁺ T Cells In Vivo. *PLoS One* **5**,

- e15328 (2010).
4. Zappasodi, R. *et al.* Rational design of anti-GITR-based combination immunotherapy. *Nat. Med.* **25**, 759–766 (2019).
 5. Dodagatta-Marri, E. *et al.* α -PD-1 therapy elevates Treg/Th balance and increases tumor cell pSmad3 that are both targeted by α -TGF β antibody to promote durable rejection and immunity in squamous cell carcinomas. *J. Immunother. Cancer* **7**, 62 (2019).
 6. Togashi, Y., Shitara, K. & Nishikawa, H. Regulatory T cells in cancer immunosuppression — implications for anticancer therapy. *Nat. Rev. Clin. Oncol.* **16**, 356–371 (2019).
 7. Kumagai, S. *et al.* The PD-1 expression balance between effector and regulatory T cells predicts the clinical efficacy of PD-1 blockade therapies. *Nat. Immunol.* (2020) doi:10.1038/s41590-020-0769-3.
 8. Kamada, T. *et al.* PD-1 + regulatory T cells amplified by PD-1 blockade promote hyperprogression of cancer. *Proc. Natl. Acad. Sci.* **116**, 9999–10008 (2019).
 9. Kusaka, Y. *et al.* Potential Role of Gr-1⁺ CD8⁺ T Lymphocytes as a Source of Interferon- γ and M1/M2 Polarization during the Acute Phase of Murine *Legionella pneumophila* Pneumonia. *J. Innate Immun.* **10**, 328–338 (2018).
 10. Matsuzaki, J. *et al.* Successful elimination of memory-type CD8⁺ T cell subsets by the administration of anti-Gr-1 monoclonal antibody in vivo. *Cell. Immunol.* **224**, 98–105 (2003).
 11. Lhuillier, C. *et al.* Impact of Exogenous Galectin-9 on Human T Cells. *J. Biol. Chem.* **290**, 16797–16811 (2015).
 12. Wu, C. *et al.* Galectin-9-CD44 Interaction Enhances Stability and Function of Adaptive Regulatory T Cells. *Immunity* **41**, 270–282 (2014).
 13. Yan, J. *et al.* Tim-3 Expression Defines Regulatory T Cells in Human Tumors. *PLoS One* **8**, e58006 (2013).
 14. Sakuishi, K. *et al.* TIM3 + FOXP3 + regulatory T cells are tissue-specific promoters of T-cell dysfunction in cancer. *Oncoimmunology* **2**, e23849 (2013).
 15. Kallies, A., Zehn, D. & Utzschneider, D. T. Precursor exhausted T cells: key to successful immunotherapy? *Nat. Rev. Immunol.* **20**, 128–136 (2020).
 16. McLane, L. M., Abdel-Hakeem, M. S. & Wherry, E. J. CD8 T Cell Exhaustion During Chronic Viral Infection and Cancer. *Annu. Rev. Immunol.* **37**, 457–495 (2019).
 17. Blank, C. U. *et al.* Defining ‘T cell exhaustion’. *Nat. Rev. Immunol.* **19**, 665–674 (2019).
 18. Wei, S. C. *et al.* Distinct Cellular Mechanisms Underlie Anti-CTLA-4 and Anti-PD-1 Checkpoint Blockade. *Cell* **170**, 1–14 (2017).
 19. Jansen, C. S. *et al.* An intra-tumoral niche maintains and differentiates stem-like CD8 T cells. *Nature* **576**, 465–470 (2019).
 20. Im, S. J. *et al.* Defining CD8⁺ T cells that provide the proliferative burst after PD-1 therapy. *Nature* **537**, 417–421 (2016).

21. Sade-Feldman, M. *et al.* Defining T Cell States Associated with Response to Checkpoint Immunotherapy in Melanoma. *Cell* **175**, 998-1013.e20 (2018).
22. Siddiqui, I. *et al.* Intratumoral Tcf1+PD-1+CD8+ T Cells with Stem-like Properties Promote Tumor Control in Response to Vaccination and Checkpoint Blockade Immunotherapy. *Immunity* **50**, 195-211.e10 (2019).
23. Li, H. *et al.* Dysfunctional CD8 T Cells Form a Proliferative, Dynamically Regulated Compartment within Human Melanoma. *Cell* **176**, 775-789.e18 (2019).
24. Miller, B. C. *et al.* Subsets of exhausted CD8+ T cells differentially mediate tumor control and respond to checkpoint blockade. *Nat. Immunol.* **20**, 326–336 (2019).
25. Coe, D. *et al.* Depletion of regulatory T cells by anti-GITR mAb as a novel mechanism for cancer immunotherapy. *Cancer Immunol. Immunother.* **59**, 1367–1377 (2010).
26. Bulliard, Y. *et al.* Activating Fc γ receptors contribute to the antitumor activities of immunoregulatory receptor-targeting antibodies. *J. Exp. Med.* **210**, 1685–1693 (2013).
27. Mahne, A. E. *et al.* Dual roles for regulatory T-cell depletion and costimulatory signaling in agonistic GITR targeting for tumor immunotherapy. *Cancer Res.* **77**, 1108–1118 (2017).
28. Gautron, A.-S. S., Dominguez-Villar, M., de Marcken, M. & Hafler, D. A. Enhanced suppressor function of TIM-3 + FoxP3 + regulatory T cells. *Eur. J. Immunol.* **44**, 2703–2711 (2014).
29. Andzinski, L. *et al.* Growing tumors induce a local STING dependent Type I IFN response in dendritic cells. *Int J Cancer* **139**, 1350–1357 (2016).
30. Lechner, M. G. *et al.* Immunogenicity of Murine Solid Tumor Models as a Defining Feature of In Vivo Behavior and Response to Immunotherapy. *J. Immunother.* **36**, 477–489 (2013).
31. Galon, J. & Bruni, D. Approaches to treat immune hot, altered and cold tumours with combination immunotherapies. *Nat. Rev. Drug Discov.* **18**, 197–218 (2019).
32. Sanmamed, M. F., Chester, C., Melero, I. & Kohrt, H. Defining the optimal murine models to investigate immune checkpoint blockers and their combination with other immunotherapies. *Ann. Oncol.* **27**, 1190–1198 (2016).
33. Mosely, S. I. S. *et al.* Rational Selection of Syngeneic Preclinical Tumor Models for Immunotherapeutic Drug Discovery. *Cancer Immunol. Res.* **5**, 29–41 (2017).
34. Zhong, W. *et al.* Comparison of the molecular and cellular phenotypes of common mouse syngeneic models with human tumors. *BMC Genomics* **21**, 1–17 (2020).

REVIEWERS' COMMENTS

Reviewer #1 (Remarks to the Author):

Point 1:

-Repeat aGal9 and GITR experiment appears sound.
-I would have preferred to see tabulated absolute numbers (i.e. compiled bar graphs), but the added data is sufficient and supports the authors' conclusions.

Point 2:

-Very well done.

Specific comments and minor point:

-All addressed to satisfaction.

The authors should be commended for all of the effort that went into improving this manuscript, particularly during challenges of the COVID19 era. This updated manuscript is now fantastic report that adds greatly to our understanding of Gal-9 in cancer.

Reviewer #2 (Remarks to the Author):

In the revised paper by Yang et al., "Galectin-9 interacts with PD-1 to regulate T cell death and is a target for cancer immunotherapy", the authors have addressed most of the comments that were raised, in most cases with the addition of new data. The paper has improved considerably.

The logic for using anti-GITR antibody instead of anti-Tim-3 or anti-PD-1 should be spelled out, as anti-PD-1 protects Tregs and promotes their Treg function and that is why the anti-GITR antibody was used. Why the combination with anti-Tim-3 was not used, because it impacts the same pathway should also be provided in the rationale for using anti-GITR in the paper.

Ca fluxes be an early indication that Tim-3 has been engaged by Galectin-9, although the authors have not used it as suggested in the initial review? Galectin induced Ca flux is a pre-requisite for Tim-3+ cells to clump and die, however, we do not know whether the death is by apoptosis.

Since the paper has 50% of the data on Tim-3, therefore it is suggested that TIM-3 should be featured in the title of the paper as well, which will increase its relevance.

Reviewer #3 (Remarks to the Author):

The authors have addressed my queries. I consider the work ready to be published.

RESPONSE TO REVIEWERS' COMMENTS

Reviewer #1 (Remarks to the Author):

Point 1:

-Repeat aGal9 and GITR experiment appears sound.

-I would have preferred to see tabulated absolute numbers (i.e. compiled bar graphs), but the added data is sufficient and supports the authors' conclusions.

Response. *We are glad the reviewer agrees that the data is sufficient to support the conclusions.*

Point 2:

-Very well done.

Specific comments and minor point:

-All addressed to satisfaction.

The authors should be commended for all of the effort that went into improving this manuscript, particularly during challenges of the COVID19 era. This updated manuscript is now fantastic report that adds greatly to our understanding of Gal-9 in cancer.

Response. *We like to thank the reviewer for the generous compliment. This is only made possible by the very helpful comments and suggestions from all the reviewers.*

Reviewer #2 (Remarks to the Author):

In the revised paper by Yang et al., "Galectin-9 interacts with PD-1 to regulate T cell death and is a target for cancer immunotherapy", the authors have addressed most of the comments that were raised, in most cases with the addition of new data. The paper has improved considerably.

Response. *We like to thank the reviewer for this positive comment. The reviewers' suggestions and comments have played a major role in our efforts to improve the manuscript.*

The logic for using anti-GITR antibody instead of anti-Tim-3 or anti-PD-1 should be spelled out, as anti-PD-1 protects Tregs and promotes their Treg function and that is why the anti-GITR antibody was used. Why the combination with anti-Tim-3 was not used, because it impacts the same pathway should also be provided in the rationale for using anti-GITR in the paper.

Response. *We like to thank the reviewer for this important suggestion. As suggested, we have added the following sentence for anti-TIM-3 (p10, para 1): "Combination of anti-Gal-9 with anti-TIM-3 has not been attempted because the two presumably act on overlapping pathways and their combination is unlikely to yield synergistic effects".*

We did evaluate anti-Gal-9/anti-PD-L1 (Supplementary Fig. 4e-h), and have added the following sentences in the Discussion section of the current version (p17, para 2): "Combination of anti-

Gal-9 with anti-PD-L1 appears to have lower antitumor therapeutic efficacy than the combination with anti-GITR, probably because PD-1 blockade, while recovers dysfunctional PD-1⁺ CD8 T cells, in the meantime also enhances PD-1⁺ Treg cell-mediated immunosuppression¹⁻³. In fact, PD-1⁺ Tregs amplified by PD-1 blockade have been shown to promote hyperprogression of cancer⁴”.

Ca fluxes be an early indication that Tim-3 has been engaged by Galectin-9, although the authors have not used it as suggested in the initial review? Galectin induced Ca flux is a prerequisite for Tim-3+ cells to clump and die, however, we do not know whether the death is by apoptosis.

Response: *Yes. We appreciate the suggestion and are aware of the very nice work by Zhu et al showing that calcium flux is an early event subsequent to Gal-9 binding to TIM-3 on the cell surface, before the induction of cell death⁵. We skipped Ca²⁺ flux and directly measured cell death for the following reasons:*

1) In T cells Ca²⁺ flux is generally believed to be associated more with cell activation than with cell death⁶.

2) The significance of Ca²⁺ flux in Gal-9-induced T cell death is controversial. Although Ca²⁺ flux precedes cell death in Gal-9-treated T cells, Jurkat T cell sublines defective in Gal-9-induced Ca²⁺ flux remain sensitive to Gal-9-induced cell death⁷, suggesting that Ca²⁺ mobilization is not required for Gal-9-induced cell death, at least in Jurkat cells.

3) Gal-9 also induces Ca²⁺ flux in parental Jurkat cells that do not express TIM-3⁷, and in TIM-3 knockout mouse Th1 cells (albeit to a lesser extent)⁵, suggesting that Gal-9-induced Ca²⁺ mobilization is not unique for TIM-3.

4) This study deals with heterogenous populations of cells with differential TIM-3/PD-1 expression and it is hard to associate Ca²⁺ flux with a particular subpopulation of cells with currently available assays.

We hope the reviewer agrees that our observation of increased Gal-9-induced death of TIM-3-expressing T cells already suggests activation of the Gal-9/TIM-3 cell death pathway, and reduction of such cell death by PD-1 co-expression is sufficient to support our conclusion that co-expressed PD-1 reduces Gal-9/TIM-3-induced T cell death.

Since the paper has 50% of the data on Tim-3, therefore it is suggested that TIM-3 should be featured in the title of the paper as well, which will increase its relevance.

Response. *This is an excellent suggestion. We have changed the title to “Galectin-9 interacts with PD-1 and TIM-3 to regulate T cell death and is a target for cancer immunotherapy”*

Reviewer #3 (Remarks to the Author):

The authors have addressed my queries. I consider the work ready to be published.

Response: *We like to thank the reviewer for this positive comment, and those to the previous version that have helped us tremendously to improve the manuscript.*

References

1. Dodagatta-Marri, E. *et al.* α -PD-1 therapy elevates Treg/Th balance and increases tumor cell pSmad3 that are both targeted by α -TGF β antibody to promote durable rejection and immunity in squamous cell carcinomas. *J. Immunother. Cancer* **7**, 62 (2019).
2. Togashi, Y., Shitara, K. & Nishikawa, H. Regulatory T cells in cancer immunosuppression — implications for anticancer therapy. *Nat. Rev. Clin. Oncol.* **16**, 356–371 (2019).
3. Kumagai, S. *et al.* The PD-1 expression balance between effector and regulatory T cells predicts the clinical efficacy of PD-1 blockade therapies. *Nat. Immunol.* (2020) doi:10.1038/s41590-020-0769-3.
4. Kamada, T. *et al.* PD-1 + regulatory T cells amplified by PD-1 blockade promote hyperprogression of cancer. *Proc. Natl. Acad. Sci.* **116**, 9999–10008 (2019).
5. Zhu, C. *et al.* The Tim-3 ligand galectin-9 negatively regulates T helper type 1 immunity. *Nat. Immunol.* **6**, 1245–1252 (2005).
6. Trebak, M. & Kinet, J.-P. Calcium signalling in T cells. *Nat. Rev. Immunol.* **19**, 154–169 (2019).
7. Lhuillier, C. *et al.* Impact of Exogenous Galectin-9 on Human T Cells. *J. Biol. Chem.* **290**, 16797–16811 (2015).